# Rapid and simultaneous detection of *Escherichia coli and Klebsiella pneumoniae:* a novel dual recombinase polymerase amplification-clustered regularly interspaced short palindromic repeats-Cas12a method

Lijian Wei,[1,2,3,4,5,6] Guangfu Pang,[1,2,3,4,5] Shihua Luo,[1,2,3,4,5] Weijie Zhou,[6] Baoyan Ren,[7] Miao Li,[8] Guijiang Wei,[1,2,3,4,5] Xuebin Li,[1,2,3,4,5] Lina Liang[1,2,3,4,5]

**ABSTRACT** *Escherichia coli* and *Klebsiella pneumoniae* are major hospital-acquired pathogens, posing severe threats to critically ill and immunocompromised patients. Their pathogenicity and virulence factors easily cause invasive infections, endangering patient health and life. This study developed a dual recombinase polymerase amplification combined with Clustered Regularly Interspaced Short Palindromic Repeats-Cas12a assay for rapid, simultaneous, specific detection of *E. coli uidA* and *K. pneumoniae rcsA* genes. The assay operated at 37℃, and the total reaction time was approximately 70 min. The analytical sensitivity for *E. coli* and *K. pneumoniae* was $5.37 \times 10^1$ copies/µL and $5.90 \times 10^1$ copies/µL, respectively. It showed excellent specificity (no cross-reactivity) and 100% concordance with PCR results in clinical sample validation. This assay overcame limitations of traditional methods, such as bacterial culture (time-consuming) and PCR (the dependency on expensive thermal cyclers), and offers advantages of rapidity, high sensitivity, simplicity, and cost-effectiveness, providing a powerful tool for rapid, accurate clinical diagnosis of *E. coli* and *K. pneumoniae* infections.

**IMPORTANCE** *Escherichia coli* and *Klebsiella pneumoniae* are the main pathogens causing hospital-acquired infections, which can lead to serious complications and pose a significant challenge to public health. Therefore, establishing rapid, sensitive, specific, and reliable detection methods for *E. coli* and *K. pneumoniae* is of great significance for promoting accurate early clinical diagnosis and guiding treatment decisions. With the increasing incidence of mixed infections, single-target nucleic acid testing can no longer meet clinical needs. In this study, the efficient recombinase polymerase amplification (RPA) isothermal amplification technique was combined with the highly sensitive Clustered Regularly Interspaced Short Palindromic Repeats (CRISPR)-Cas12a detection system to successfully develop a duplex RPA-CRISPR-Cas12a method. This system can specifically and simultaneously identify *E. coli* (targeting the *uidA* gene) and *K. pneumoniae* (targeting the *rcsA* gene) in a single detection process.

**KEYWORDS** dual RPA, CRISPR/Cas12a, *Escherichia coli*, *Klebsiella pneumoniae*

$E$scherichia coli and *Klebsiella pneumoniae* are recognized as primary etiological agents of healthcare-associated infections, posing significant clinical challenges (1, 2). As opportunistic pathogens, *E. coli* and *K. pneumoniae* primarily threaten immunocompromised patients and those with pre-existing comorbidities (3). The virulence mechanisms of these pathogens often lead to severe systemic infections, which significantly increase patient morbidity and mortality. These infections manifest as a range of conditions,

Address correspondence to Lina Liang, 13481684467@ymun.edu.cn, Xuebin Li, 00025@ymun.edu.cn, or Guijiang Wei, 1581@ymun.edu.cn.

The authors declare no conflict of interest.

including respiratory tract infections, meningitis, sepsis, and stroke-associated infections (4, 5). The emergence of hypervirulent and drug-resistant strains further exacerbates the difficulties in clinical management and hinders the implementation of effective infection control measures.

Conventional detection methods for *E. coli* and *K. pneumoniae* predominantly rely on cultural techniques (6). While conventional culture methods are widely adopted in resource-limited settings owing to their low analytical cost and operational simplicity, they require extended incubation periods (typically 24–72 h). This prolonged time-frame poses inherent risks of false-positive and false-negative results, constraining their applicability in time-sensitive diagnostic contexts (7). Despite their enhanced analytical sensitivity and specificity, molecular methods such as polymerase chain reaction (PCR) remain constrained by lengthy processing workflows, dependence on costly thermocycling equipment, and the requirement for specialized personnel—factors that restrict their applicability in point-of-care settings requiring rapid diagnostics (8). Consequently, the development of novel detection platforms with accelerated turnaround time, operational simplicity, analytical accuracy, and high sensitivity is imperative to facilitate early diagnosis, implement timely interventions, and mitigate pathogen transmission.

In recent years, the Clustered Regularly Interspaced Short Palindromic Repeats-Cas (CRISPR-Cas) system has emerged as a significant breakthrough in the realm of genetic engineering, continuously expanding the frontiers of molecular diagnostics research and its practical applications, and has become one of the most critical tools propelling advancements in this domain (9, 10). Comprising nuclease-active Cas proteins and guide RNA, this system enables the precise recognition and cleavage of specific DNA or RNA targets (11, 12). Specifically, when guided by CRISPR RNA (crRNA), CRISPR-Cas12a can cleave double-stranded DNA (dsDNA) containing a thymidine (T)-rich prototypical spacer sequence flanking the protospacer adjacent motif (PAM), which generates sticky ends (13). In contrast, Cas12a recognizes and cleaves single-stranded DNA (ssDNA) targets independently of PAM sequences. A key functional characteristic of Cas12a is its trans-cleavage activity: upon binding to its target dsDNA, it gains the ability to non-specifically degrade ssDNA with high catalytic efficiency (14, 15). This unique activity has been successfully harnessed for the development of highly sensitive biosensors capable of detecting trace amounts of nucleic acid targets (16). For instance, Ye et al. developed a CRISPR-Cas12a-based SERS nanosensor for highly sensitive detection of HPV DNA, achieving a detection limit as low as 209 copies/µL (17). This technology is suitable for rapid point-of-care diagnostics in resource-limited settings. Similarly, Jiang et al. developed a rapid detection platform based on multiplex recombinase polymerase amplification (RPA) combined with CRISPR-Cas12a for detecting *Clostridium difficile* toxins A and B, which also exhibits high sensitivity and specificity for rapid on-site detection (18).

The sensitivity of nucleic acid testing typically requires pre-amplification of the target sequence. However, conventional PCR is limited by its requirement for precise thermal cycling, extended processing times, and dependence on specialized instrumentation, restricting its utility in point-of-care or resource-limited settings (19). RPA, a rapid isothermal amplification technique, addresses these limitations by enabling efficient target amplification at a constant temperature (typically 37°C–42°C) within 15–30 min. This feature not only streamlines workflow but also reduces reliance on complex equipment, making RPA particularly suitable for decentralized diagnostic applications (20). Significantly, the integration of the operational simplicity of RPA and the high specificity and signal amplification capacity of CRISPR-Cas systems, particularly Cas12a, forms a highly effective detection strategy. The integration of RPA with CRISPR-Cas12a thus forms a robust platform for rapid molecular diagnostics and has been successfully applied to detect diverse pathogens. For instance, Chen et al. developed a one-pot RPA-CRISPR-Cas12a-electrochemical biosensor for ultrasensitive detection of *Salmonella* (21). Similarly, Bao et al. described an RPA-CRISPR-Cas12a assay coupled with lateral flow detection, offering a portable, rapid, and sensitive readout (22). Furthermore, Xie et

al. introduced MIIND-DMF, an automated microfluidic platform integrating RPA-CRISPR technology for multiplex bacterial detection (23).

Building on this foundation, we developed an integrated dual RPA-CRISPR-Cas12a assay for the simultaneous, sensitive, and specific detection of *E. coli* and *K. pneumoniae*. The assay targets the *uidA* gene, a well-established marker for *E. coli* (24, 25), and the *rcsA* gene, which regulates capsular polysaccharide synthesis in *K. pneumoniae* (26). To our knowledge, this work represents the first report of a combined RPA-CRISPR-Cas12a method for simultaneously detecting these two pathogens via these targets. This integrated approach offers several distinct advantages: (i) RPA enables rapid, isothermal amplification with minimal equipment requirements; (ii) CRISPR-Cas12a ensures high specificity through crRNA-guided recognition; (iii) the dual-target design increases diagnostic throughput; and (iv) Cas12a's trans-cleavage activity provides a sensitive and universal signal readout. The resulting assay is simple, rapid, and cost-effective, offering a promising tool for the accurate identification of these clinically important pathogens.

## MATERIALS AND METHODS

### Standard strains and DNA extraction

Details of the pathogenic bacterial strains employed in this study are documented in Table S1. Genomic DNA extraction was conducted using a commercial bacterial DNA extraction kit (TianGen, Beijing), adhering to the manufacturer's protocols. Extracted DNA was quantified spectrophotometrically, with samples meeting quality control criteria (OD260/280 = 1.8–2.0) retained for analysis. Residual DNA aliquots were cryopreserved at −20℃.

### DNA extraction process

Genomic DNA was extracted from clinical bacterial isolates using the TIANamp Bacteria DNA Kit according to the manufacturer's protocol. Briefly, 1 mL–5 mL of bacterial culture was centrifuged to obtain a pellet. The pellet was resuspended in Buffer GA, followed by the addition of Proteinase K and Buffer GB for cell lysis at 70℃ for 10 min. After adding absolute ethanol, the lysate was transferred to a spin column for DNA binding. The column was washed twice with Buffer GD and Buffer PW, respectively. Pure genomic DNA was finally eluted in 50 µL–200 µL of Elution Buffer TE and stored at −20℃. The concentration and purity of the DNA were measured using a NanoDrop spectrophotometer. The extracted DNA was used as the template for RPA-CRISPR-Cas12a reaction.

To minimize the risk of contamination, we will strictly follow the following operations: (i) Zoned operation: All experimental steps will be carried out in physically separated spaces. Reagent preparation and sample processing are carried out in the clean preparation area, RPA is conducted in the amplification area, and the opening of the amplification product caps and CRISPR detection are completed in the independent product analysis area, following a one-way workflow. (ii) Anti-contamination operation: Use pipette tips with filter cores throughout the process. Laboratory personnel should wear gloves and change them frequently. After each experimental batch, thoroughly clean the working area with a nucleic acid remover. (iii) Negative control setup: We set up a template-free control (NTC) in each experimental batch to monitor whether there is any contamination of amplification products from reagents or the environment throughout the process.

### RPA primer and crRNA design

The conserved sequences of the *E. coli uidA* gene (KT311783.1) and *K. pneumoniae rcsA* gene (AY059955.1) were obtained from the NCBI website. Following the primer design guidelines provided by the TwistAmp Basic Kit (TwistDx, USA), primer sequences were designed employing the NCBI Primer-BLAST tool, and primer specificity was verified using "primer-blast." For this conserved sequence, we designed six primer pairs for the

*rcsA* gene (*rcsA* -F1/R1, *rcsA*-F2/R2, *rcsA*-F3/R3, *rcsA*-F4/R4, *rcsA*-F5/R5, *rcsA*-F6/R6) and four primer pairs for the *uidA* gene (*uidA*-F1/R1, *uidA*-F2/R2, *uidA*-F3/R3, *uidA*-F4/R4). Analysis of RPA products was performed by evaluation of optimal primers via 2% agarose gel electrophoresis. Following primer selection, corresponding crRNAs (*uidA*-crRNA1/2, *rcsA*-crRNA1/2) were designed using CRISPR online tools. Based on the CRISPR-Cas12a cleavage mechanism, 5′-labeled FAM and 3′-labeled BHQ1 ssDNA reporter genes were designed for fluorescent detection. The corresponding crRNA and ssDNA sequences are provided in Table S2.

## Establishment of a dual RPA-CRISPR-Cas12a detection method

To achieve efficient synchronous detection of *E. coli* and *K. pneumoniae*, we have developed a detection system based on dual RPA and CRISPR-Cas12a. During the nucleic acid amplification stage, we designed primers targeting the *E. coli* conserved gene *uidA* and the *K. pneumoniae*-specific gene *rcsA,* respectively, and simultaneously amplified the two targets in the same RPA reaction tube. However, due to the non-specific activity of the activated Cas12a protein in cutting single-stranded DNA, to prevent crRNAs targeting different pathogens from interfering with each other in the same reaction system, in the CRISPR-Cas12a detection step, we amplified the RPA product. Incubate with specific crRNA and Cas12a protein, respectively, in independent reaction tubes. Ultimately, by real-time monitoring of the fluorescence intensity of each reaction tube, specific recognition and determination of the two pathogens were achieved.

## Dual-RPA system

The Dual-RPA reaction in this study aims to simultaneously detect two targets (the *rcsA* gene of *K. pneumoniae* and the *uidA* gene of *E. coli*); thus, a total of four primers targeting both targets are included in a single 10 µL reaction system. To save reagent costs, we adopt a strategy of first preparing a concentrated mother liquor with multiple reaction parts and then evenly dividing it. The specific steps are as follows: (i) Preparation and aliquoting of 5 reaction parts of premixed mother liquor: Take a complete Twist-Amp Basic freeze-dried RPA enzyme sphere. Use a micropipette to add the following components to the enzyme ball to prepare a premixed stock solution sufficient for five reactions: 10× TwistAmp Basic buffer: 29.5 µL, nuclease-free water: 4.4 µL, *K. pneumoniae* forward primer (10 µM): 2.4 µL, *K. pneumoniae* reverse primer (10 µM): 2.4 µL, *E. coli* forward primer (10 µM): 2.4 µL, *E. coli* reverse primer (10 µM): 2.4 µL, total volume: 43.5 µL. Repeatedly pipette or briefly vortex with a pipette to ensure that the freeze-dried enzyme spheres are completely dissolved and well mixed. The above 43.5 µL premixed mother liquor was evenly aliquoted into five independent reaction tubes, with each tube receiving 8.7 µL. (ii) Completion of a single reaction (10 µL system): Add the following in sequence to the reaction tube that already contains 8.7 µL of premix: *K. pneumoniae* DNA template: 0.4 µL, *E. coli* DNA template: 0.4 µL. (Note: In the negative control, both templates should be replaced with equal volumes of nuclease-free water.) Gently pipette and mix well. Finally, add 0.5 µL of 280 mM magnesium acetate (MgOAc) solution to initiate the reaction and immediately centrifuge briefly. Reaction procedure: Immediately place the reaction tube in a constant-temperature metal bath at 37°C and incubate for 25 min. The Dual-RPA reaction system is shown in Table S5.

## CRISPR-Cas12a cleavage assay

For the *K. pneumoniae rcsA* gene and the *E. coli uidA* gene, we designed two fluorescent detection systems. *K. pneumoniae* (*rcsA*) testing system: The 20 µL reaction system contains 2 µL of 10× NEBuffer r2.1, 2 µL of amplification products, 500 nM fluorescent reporter molecule (ssDNA, 5′FAM-TTATT-BHQ13′), 200 nM LbCas12a, and 200 nM crRNA, which is replenished to 20 µL with nuclease-free water. *E. coli* (*uidA*) detection system: The 20 µL reaction system contains 2 µL of 10× NEBuffer r2.1, 2 µL of amplification products, 600 nM fluorescent reporter molecule (ssDNA, 5′FAM-TTATT-BHQ13′), 300

nM LbCas12a and 300 nM crRNA, which is replenished to 20 µL with nuclease-free water. Following dual RPA, the amplified products were transferred to their respective fluorescent detection systems. The cleavage process was tracked in real time using a qPCR instrument. The reactions were carried out at 37℃ for 30 min, with the fluorescence intensity recorded every 60 s. Endpoint fluorescence data were processed using GraphPad Prism 10 (GraphPad Software, Inc., USA). Fluorescence can also be excited under ultraviolet (UV) light, and images can be taken with a smartphone.

## Optimization of a dual-RPA-CRISPR-Cas12 method

The objective of optimizing the dual RPA system was to achieve efficient, simultaneous amplification of both *uidA* and rcsA target genes. The amplification performance was evaluated based on three primary criteria: (i) the intensity of the amplification band on agarose gel, indicating yield; (ii) the specificity, indicated by a single, sharp band at the expected molecular size for each target; and (iii) the clarity of the electrophoretic background, with minimal smearing or non-specific products. To achieve this, we first investigated the potential competitive inhibition between the two primer pairs by conducting gradient tests on the concentration of each primer set within the range of 240 nM to 560 nM, aiming to determine the optimal formulation ratio. Subsequently, different reaction temperatures (34℃, 37℃, 40℃, and 43℃) and reaction times (10 min, 15 min, 20 min, 25 min, and 30 min) were tested. Reaction outcomes were evaluated using 2% agarose gel electrophoresis to determine the optimal conditions for dual RPA.

Additionally, to ensure the cleavage efficiency of the CRISPR-Cas12a system, we optimized the concentrations of Cas12a, crRNA, and ssDNA. The total volume of the CRISPR-Cas12a system was consistently maintained at 20 µL. Select the optimal concentration of each component based on the criterion for generating the highest fluorescence intensity, which corresponds to the maximum cleavage activity. We tested six different Cas12a concentrations (100 nM, 200 nM, 300 nM, 400 nM, 500 nM, and 600 nM) and selected the optimal Cas12a concentration based on fluorescence intensity. We tested six different crRNA concentrations (100 nM, 200 nM, 300 nM, 400 nM, 500 nM, and 600 nM) and selected the optimal crRNA based on fluorescence intensity. Fluorescent reporter probe concentration is critical for detecting fluorescence signals in the CRISPR-Cas12a reaction system. While maintaining other reaction conditions constant, fluorescent reporter probes at seven different concentrations (100 nM, 200 nM, 300 nM, 400 nM, 500 nM, 600 nM, and 700 nM) were added to the optimized reaction system. Fluorescence data were recorded using a qPCR instrument. All experiments were repeated three times.

## Validation of sensitivity and specificity in a dual RPA-CRISPR-Cas12 method

To ensure the reliability of sensitivity detection results, we designed and synthesized *E. coli uidA* and *K. pneumoniae rcsA* plasmids as templates. Using enzyme-free water as the diluent, the *uidA* plasmid was 10-fold serially diluted from $5.37 \times 10^{10}$ copies/µL to $5.37 \times 10^{-1}$ copies/µL; the *rcsA* plasmid was 10-fold serially diluted from $5.9 \times 10^{10}$ copies/µL to $5.9 \times 10^{-1}$ copies/µL. Sensitivity validation was performed using an optimized dual RPA-CRISPR-Cas12 system. Subsequently, we validated the specificity of this detection system by testing other clinically common pathogens, including *Pseudomonas aeruginosa*, *Staphylococcus aureus*, *Streptococcus pneumoniae*, *Acinetobacter baumannii*, *Enterococcus faecalis*, *Enterococcus faecium*, *Haemophilus influenzae*, *Enterobacter cloacae*, *Stenotrophomonas maltophilia*, and *Moraxella catarrhalis*. All experiments were repeated three times.

## Dual-RPA-CRISPR-Cas12 reaction system disturbance resistance experiment

A qualified detection system should possess robust interference resistance. In a dual RPA system, interference resistance validation is performed by altering the template while keeping other components constant. Experimental Groups: To evaluate the detection

system's interference resistance, *Staphylococcus aureus*, *Pseudomonas aeruginosa*, and *Enterococcus faecalis* were selected as interfering bacteria. The experiment established the following four template groups: (i) target bacteria group: *E. coli* + *K. pneumoniae*; (ii) interfering bacteria group: Target bacteria group (*E. coli* + *K. pneumoniae*) + one interfering bacteria (*S. aureus*, *P. aeruginosa*, or *E. faecalis*); (iii) mixed interfering bacteria group: A mixture of three interfering bacteria (*S. aureus*, *P. aeruginosa*, *E. faecalis*); (iv) negative control group: Nuclease-free water. The number of interfering bacteria added should be equal to that of the target bacteria.

## Repeatability of the dual RPA-CRISPR-Cas12 method

Five independent replicate reactions were performed using a standard plasmid at a medium concentration of $10^4$ copies to validate the reproducibility of the dual RPA-CRISPR-Cas12 detection system. The fluorescence values at the reaction endpoint were used to calculate the mean, standard deviation, and relative coefficient of variation. The relative coefficient of variation across the five independent replicates was less than 10%, indicating that the constructed CRISPR-Cas12a-RPA detection system exhibits good reproducibility.

## Clinical sample testing

To evaluate the clinical applicability of this method, a dual RPA-CRISPR-Cas12 assay was used to analyze 30 specimens infected with *E. coli*, 30 specimens infected with *K. pneumoniae*, and 20 negative controls from non-*E. coli* and non-*K. pneumoniae* infections. We collected original clinical specimens (including blood, urine, and sputum, etc.) from the Affiliated Hospital of Youjiang Medical University for Nationalities. All these specimens underwent standard clinical microbiological cultures to confirm the presence of pathogens (*E. coli*, *K. pneumoniae*) and to isolate purified clinical strains. Genomic DNA was extracted from the purified clinical strains for the subsequent detection of RPA-CRISPR-Cas12a. The agreement between the dual RPA-CRISPR-Cas12a assay and the PCR method was compared for clinical sample detection results. The PCR system comprised a 50 µL reaction volume with the following components: 25 µL Es Tap MasterMix dye-free, 10 µM forward and reverse primers each, 2 µL template, and ddH$_2$O to a final volume of 50 µL. The PCR amplification began with an initial denaturation step at 94°C for 2 min. This was followed by 28 cycles, each comprising 30 s of denaturation at 94°C, 30 s of annealing at 59°C, and 30 s of extension at 72°C. The process concluded with a final extension step at 72°C for 2 min. The resulting PCR products were then analyzed by electrophoresis on a 2% agarose gel.

## Data analysis

Fluorescence detection results were analyzed using GraphPad Prism 10 software. For pairwise comparisons, a two-tailed Student's *t*-test was applied; for multiple group comparisons, a one-way analysis was performed. Differences were considered statistically significant at $P < 0.05$.

## RESULTS

### Dual RPA-CRISPR-Cas12 workflow for simultaneous detection of *E. coli* and *K. pneumoniae*

The workflow diagram for the dual RPA-CRISPR-Cas12 simultaneous detection of *E. coli* and *K. pneumoniae* is shown in Fig. 1. This study established a rapid pathogen detection system based on dual RPA-CRISPR-Cas12a, achieving precise detection by simultaneously targeting the *E. coli* conserved marker *uidA* and *K. pneumoniae*-specific gene *rcsA*. The specific procedure is as follows: First, genomic DNA is extracted from clinical samples as a template (15 min). Subsequently, dual RPA rapid nucleic acid amplification is performed, simultaneously amplifying the *rcsA* and *uidA* target sequences at 37°C using

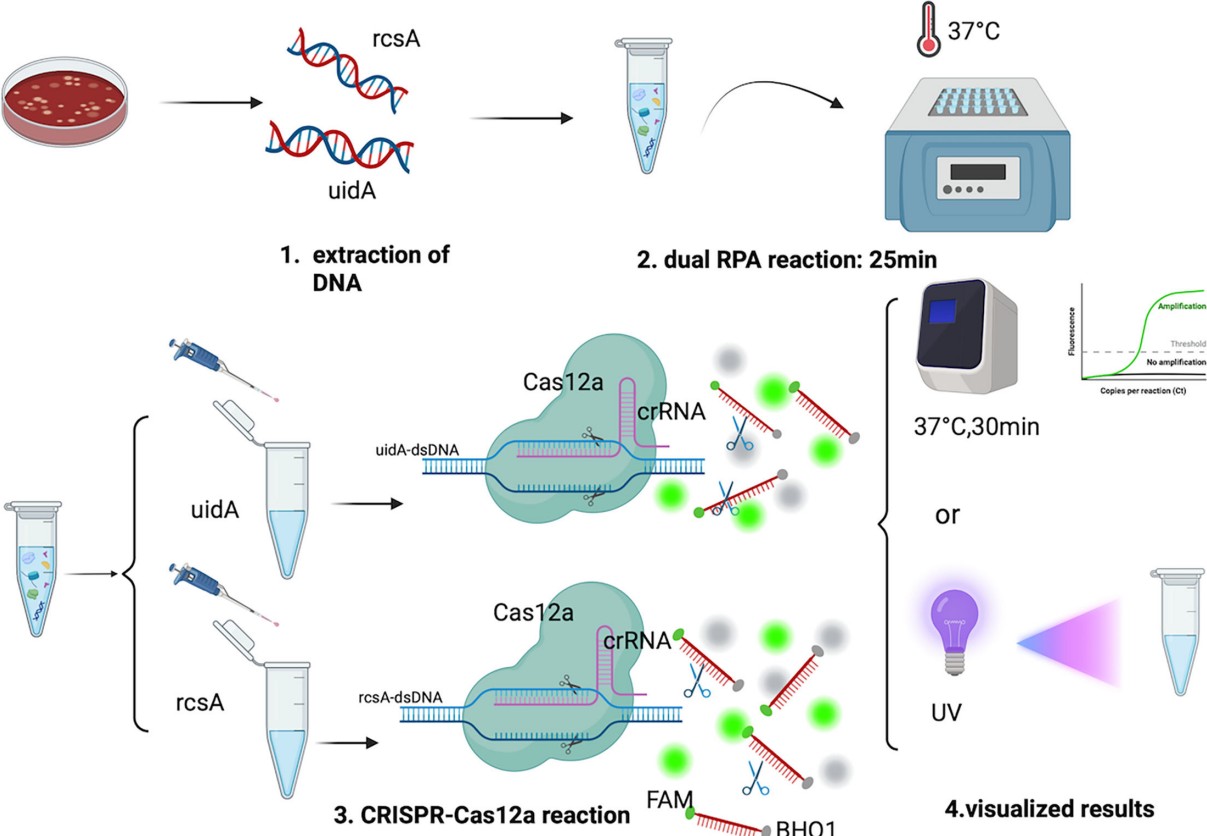

**FIG 1** Workflow for simultaneous detection of *K. pneumoniae* and *E. coli* using dual RPA-CRISPR-Cas12. 1. Nucleic acid extraction: Bacterial DNA was extracted using a nucleic acid extraction kit, with the entire process taking 15 min. 2. Dual RPA reaction: Simultaneous amplification of the *rcsA* gene from *K. pneumoniae* and the *uidA* gene from *E. coli* was performed at 37°C for 25 min. 3. CRISPR-Cas12a reaction: The amplified RPA products were added to the corresponding CRISPR-Cas12a system. After crRNA recognizes the homologous complementary sequence of the target, it guides Cas12a to recognize and cleave the target double-stranded DNA. Simultaneously, the trans-cleavage activity of Cas12a is activated, non-specifically cleaving the surrounding fluorescent probe (FAM-BHQ1). 4. Result interpretation: Results can be analyzed by recording endpoint fluorescence values using a PCR instrument or visually assessed under UV light.

specific primers (25 min). Following amplification, the products are transferred to their respective CRISPR-Cas12a fluorescent detection systems. The crRNA identifies the target DNA through complementary base pairing, activating the dual cleavage activity of the Cas12a protein. By leveraging its cis-cleavage activity, target DNA strands are cleaved at specific sites downstream of the PAM sequence. Concurrently, trans-cleavage activity triggers non-specific cleavage of the 5′-FAM/3′-BHQ1-labeled single-stranded fluorescent reporter probe. Results are visualized via real-time fluorescent PCR or UV excitation. This integrated workflow enables dual-target detection within 70 min, delivering both high sensitivity and specificity.

## Evaluation of RPA primers and crRNA

To achieve rapid and high-yield amplification of target sequences, well-designed primers are essential. For the *E. coli*-specific gene *uidA*, we designed four primer pairs; for the *K. pneumoniae*-specific gene *rcsA*, six primer pairs were designed. Detailed sequence information is provided in Table S2. All candidate primer pairs were screened using a standard RPA reaction system. Based on 2% agarose gel electrophoresis analysis, the primer pairs yielding clear amplification bands of correct size, without nonspecific bands or smearing, were selected as the optimal primer pairs. As shown in Fig. 2A and B, all primer pairs successfully amplified the target fragment. The *uidA*-F4/R4 and *rcsA*-F5/R5

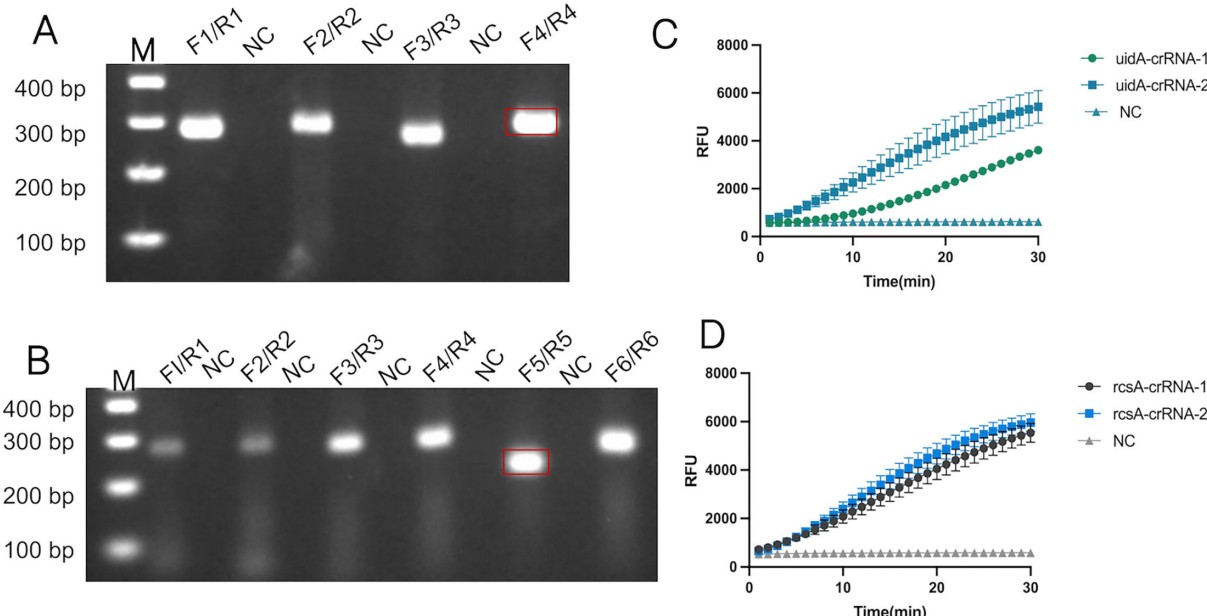

**FIG 2** Evaluation of RPA Primers and crRNA. (A) Evaluation of primers for the *E. coli uidA* gene. Four primer pairs were designed for the *uidA* gene: *uidA*-F1/R1, F2/R2, F3/R3, and F4/R4. (B) Evaluation of *K. pneumoniae rcsA* gene primers. Six primer pairs were designed for the *rcsA* gene: *rcsA*-F1/R1, F2/R2, F3/R3, F4/R4, F5/R5, and F6/R6. RPA products were analyzed by 2% agarose gel electrophoresis. (C) Evaluation of *uidA*-crRNA. (D) Evaluation of *rcsA*-crRNA. The results show that the optimal primers and crRNA are *uidA*-F4/R4, *uidA*-crRNA2; *rcsA*-F5/R5, *rcsA*-crRNA2. M, marker. NC, negative control.

primer pairs produced distinct bands without dimers or hairpin structures. Consequently, both primer pairs demonstrating optimal performance were advanced for downstream experimental workflows. Since crRNA is the core component of the CRISPR-Cas12a cleavage reaction, an appropriate crRNA is crucial for enhancing CRISPR-Cas12a cleavage efficiency. We designed two crRNAs, respectively, targeting the *uidA* and the *rcsA* genes (*uidA*-crRNA1/crRNA2 and *rcsA*-crRNA1/crRNA2), for use in subsequent CRISPR-Cas12a reactions. The design of crRNA follows its fundamental principle: each crRNA contains a conserved skeleton sequence (21 nt) for binding to the Cas12a protein and a variable interval sequence (20 nt). This spacer sequence binds to a specific homologous sequence on the target DNA (i.e., the *uidA* or *rcsA* gene) through the principle of base complementary pairing, thereby guiding the Cas12a protein to precisely locate at the target site and cleave the target DNA and the single-stranded DNA fluorescent reporter probe in the system. In the design process, we ensured that the spacer sequence of each crRNA had a high degree of gene specificity and targeted the conserved regions of the genes to maximize the reliability and sensitivity of the detection. Subsequently, we evaluated the cutting efficiency of each crRNA through *in vitro* CRISPR-Cas12a fluorescence detection experiments. As shown in Fig. 2C and D, compared with the negative control, all four crRNAs were able to successfully guide Cas12a to cut the fluorescence reporter probe and generate fluorescence signals. However, the cutting reactions guided by *uidA*-crRNA2 and *rcsA*-crRNA2 exhibited significantly higher fluorescence signal intensity and initial reaction rate, indicating that they have higher cutting efficiency. Therefore, based on their superior cutting dynamics performance, we selected *uidA*-crRNA2 and *rcsA*-crRNA2 for all subsequent experiments.

## Optimization of the dual RPA-CRISPR-Cas12 method

The concentration of primers, RPA time, and RPA temperature are all critical factors affecting RPA efficiency. We optimized the primer concentration, RPA time, and RPA reaction temperature to ensure efficient RPA. Clear and distinct amplification bands were obtained when the *uidA* primer concentration was 480 nM and the *rcsA* primer

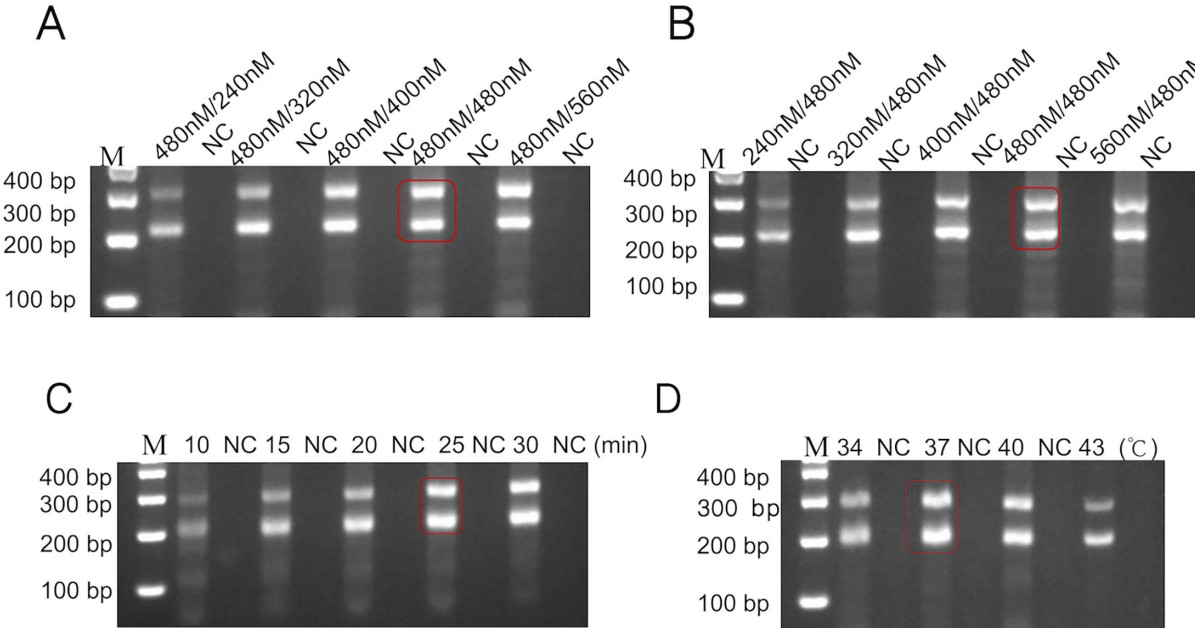

**FIG 3** Optimization of the dual RPA system. (A) Optimization of *E. coli* primer concentration. With *K. pneumoniae* primer concentration fixed at 480 nM, *E. coli* primer concentration was optimized. An *E. coli* primer concentration gradient ranging from 240 nM to 560 nM was tested to determine the optimal concentration. (B) Optimization of *K. pneumoniae* primer concentration. A similar approach was employed to optimize the *K. pneumoniae* primer concentration. Amplification products were analyzed by 2% agarose gel electrophoresis. Based on experimental results, the optimal ratio of *E. coli* primers to *K. pneumoniae* primers was 1:1 (480 nM:480 nM). Subsequent reactions were performed using this optimal primer concentration. (C) Optimization of the reaction time for dual RPA. (D) Optimization of the reaction temperature for dual RPA. After optimization, the dual RPA system can achieve the best amplification effect after reacting at 37°C for 25 min. In the figure, M, Marker. NC, negative control.

concentration was 480 nM (Fig. 3A and B). The optimal concentration of the *uidA* and *rcsA* primer pair was determined to be 480 nM and used for subsequent experiments. Subsequently, we designed tests with varying amplification times (10 min, 15 min, 20 min, 25 min, and 30 min) and reaction temperatures (34°C, 37°C, 40°C, and 43°C). Results shown in Fig. 3C and D indicate that a distinct amplification band is achieved at 25 min, while 37°C yields the highest amplification efficiency. Considering cost-effectiveness, rapid nucleic acid amplification, and amplification efficiency, the optimal dual RPA system was determined to be: *uidA* primer concentration of 480 nM; *rcsA* primer concentration of 480 nM; amplification time of 25 min; and reaction temperature of 37°C, yielding the best amplification results.

In CRISPR-Cas12a cleavage reactions, the concentrations of Cas12a, crRNA, and ssDNA are all critical factors affecting cleavage efficiency. We optimized the CRISPR-Cas12a detection systems for *E. coli* and *K. pneumoniae*, respectively. Figures 4 and 5 show the optimization results of the CRISPR-Cas12a systems for *E. coli* and *K. pneumoniae*. Figures 4A and 5A show the CRISPR-Cas12a reaction component deletion test. The results indicate that a significant fluorescence signal can only be produced when all components, including Cas12a, crRNA, ssDNA, and RPA products, are present. No matter which component is missing, no obvious fluorescence signal can be generated. More importantly, we have also added the signal-to-noise ratio (S/N) as an optimization indicator (line graph), which can more scientifically evaluate the detection performance at different concentrations. Finally, the optimal concentration should be selected based on a comprehensive consideration of cost-effectiveness. We made an intuitive and quantitative direct comparison between the baseline (conditions without changing any factors) and the results after each single-factor optimization.

The results in Fig. 4 demonstrate that, in the *E. coli* CRISPR-Cas12a system, the concentration of Cas12a (*uidA*) was set at 300 nM, the concentration of *uidA*-crRNA

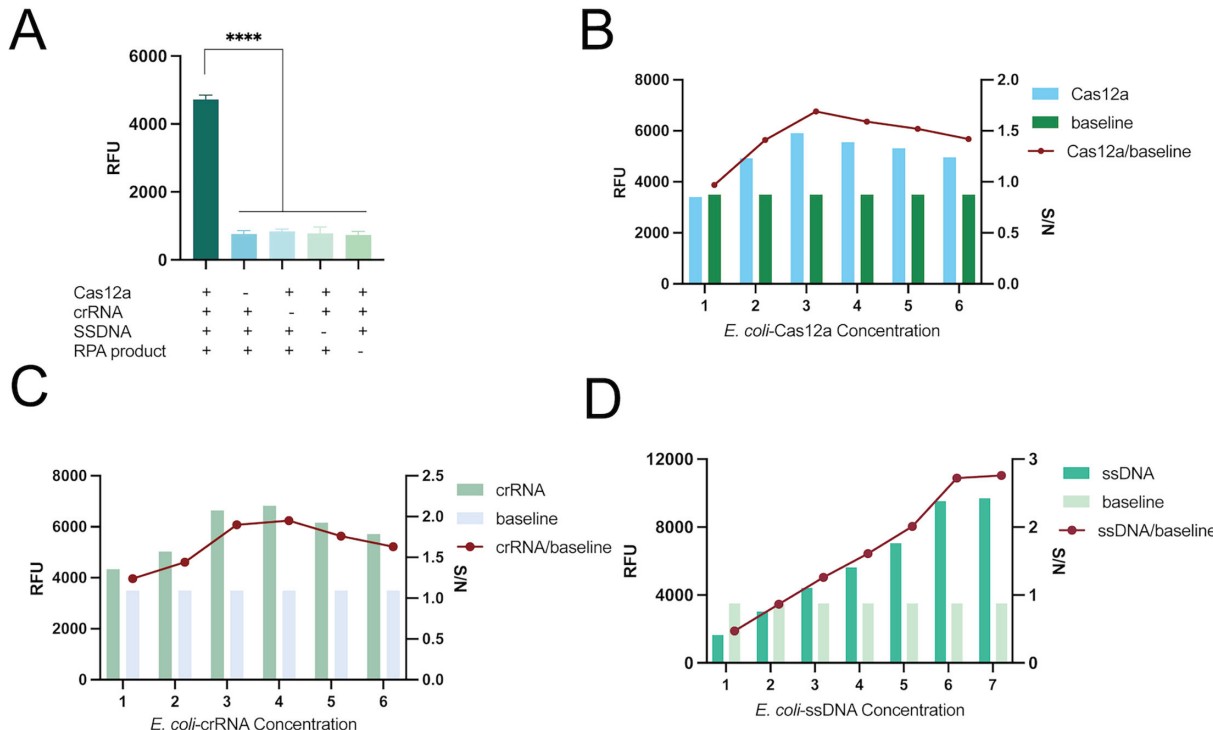

**FIG 4** Optimization of the *E. coli* CRISPR-Cas12a trans-cleavage system: (A) Verification of the dependence of key reaction components. Concentration points 1 to 7 in the subgraph (B–D) correspond respectively to 100, 200, 300, 400, 500, 600, and 700 nM. (B) Cas12a concentration optimization, with endpoint fluorescence intensity being strongest at 300 nM; (C) crRNA concentration optimization, where no significant difference in endpoint fluorescence intensity was observed between 300 nM and 400 nM crRNA (*P* > 0.05), so 300 nM was selected as the optimized concentration; (D) Optimization of ssDNA reporter concentration, with results showing the optimal ssDNA concentration was 600 nM. Baseline, conditions without changing any factors; S/N, that is, the RFU of the experimental group/baseline RFU; NC, negative control. ****, *P* < 0.0001.

was determined to be 300 nM (as there was no statistically significant difference in the resulting fluorescence intensity between 300 nM and 400 nM, *P* > 0.05), and the highest fluorescence intensity was achieved when the concentration of ssDNA (*uidA*) was 600 nM (with no statistically significant difference in fluorescence intensity observed between 600 nM and 700 nM, *P* > 0.05). Therefore, the optimal equilibrium conditions we recommend are: Cas12a: 200 nM, crRNA: 300 nM, ssDNA: 600 nM. This condition is a prudent choice based on the results of single-factor experiments, ensuring high detection efficiency (high S/N ratio) while taking into account the controllability of practical application costs. As shown in Fig. 5, for the *K. pneumoniae* CRISPR-Cas12a system, the concentrations of Cas12a (*rcsA*) and *rcsA*-crRNA were both confirmed to be 200 nM (and no statistically significant difference in fluorescence intensity was detected for *rcsA*-crRNA between 200 nM and 300 nM, *P* > 0.05); specifically, the concentration gradient assay of ssDNA (*rcsA*) indicated that a concentration of 500 nM could generate sufficiently stable fluorescence signals, which meet the requirements of subsequent experiments. The concentration of the reporting matrix is positively correlated with the signal and S/N ratio within a certain range, but after reaching a certain concentration, the gain may tend to level off, and cost-effectiveness needs to be considered. We selected 500 nM as the working concentration for subsequent experiments, as it not only fulfills the predefined experimental objectives but also offers a cost-effectiveness advantage.

## Sensitivity and specificity of the dual RPA-CRISPR-Cas12 method

A good detection method should have good sensitivity and specificity. To rigorously examine the target-specific discrimination capacity of the dual RPA-integrated CRISPR-Cas12a detection platform. We conducted tests on common clinical pathogenic bacteria

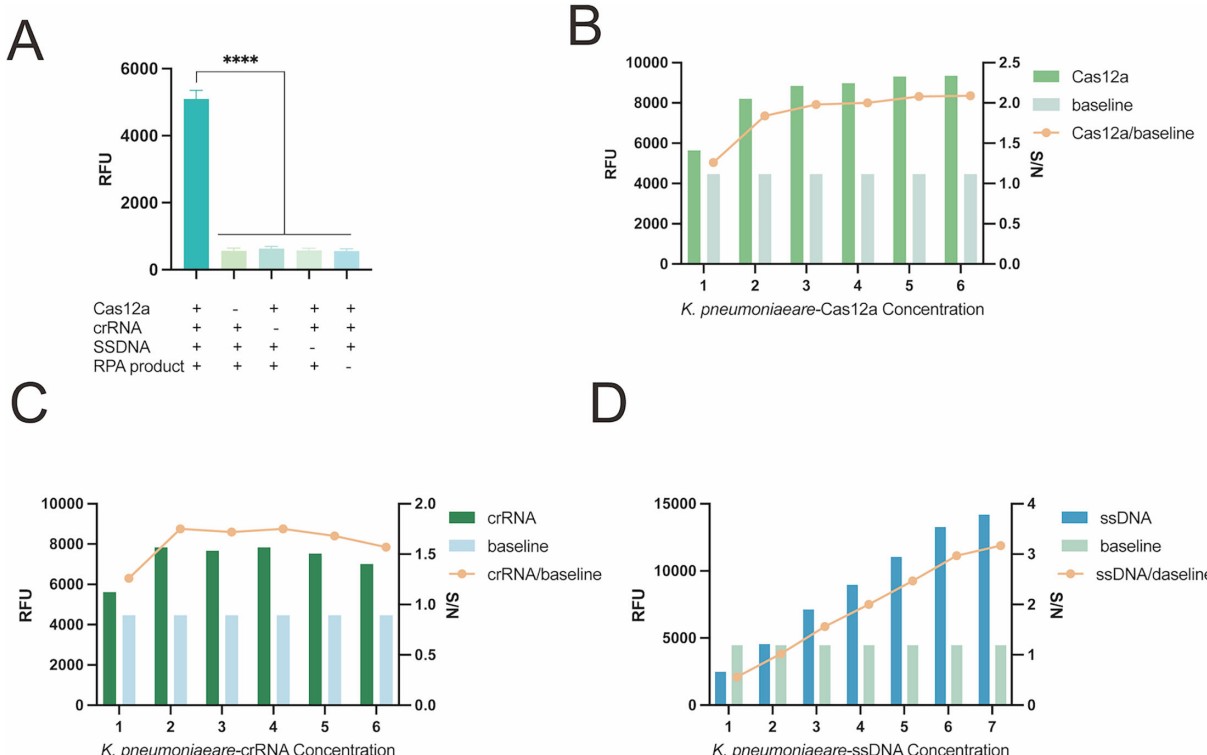

**FIG 5** Optimization of the *K. pneumoniae*-CRISPR-Cas12a trans-cleavage system. (A) Verification of the dependence of key reaction components. Concentration points 1 to 7 in the subgraph (B–D) correspond respectively to 100, 200, 300, 400, 500, 600, and 700 nM. (B) Cas12a concentration optimization: endpoint fluorescence intensity showed no significant difference between 200 nM and 300 nM ($P > 0.05$). (C) crRNA concentration optimization: no significant difference in endpoint fluorescence intensity was observed between 200 nM and 300 nM crRNA ($P > 0.05$), so 200 nM was selected as the optimized concentration. (D) Optimization of ssDNA reporter concentration. Baseline, conditions without changing any factors; S/N, that is, the RFU of the experimental group/baseline RFU; NC, negative control. ****, $P < 0.0001$.

such as *Escherichia coli*, *Klebsiella pneumoniae*, *Pseudomonas aeruginosa*, *Staphylococcus aureus*, *Streptococcus pneumoniae*, *Acinetobacter baumannii*, *Enterococcus faecalis*, *Enterococcus faecium*, *Haemophilus*, *Enterobacter cloacae*, *Stenotrophomonas maltophilia*, and *Moraxella catarrhalis*. As shown in Fig. 6A and B, only *E. coli* and *K. pneumoniae* exhibited obvious fluorescence signals, indicating that the system did not cross-react with other strains, thereby confirming its good specificity. To evaluate the sensitivity of the dual RPA-CRISPR-Cas12a system, we conducted gradient dilution experiments on *rcsA* and *uidA* genomic DNA, with concentrations ranging from $10^{10}$ to $10^{-1}$ copies/μL. As shown in Fig. 6C, robust fluorescence signals remained detectable at target concentrations as low as $10^{1}$ copies/μL (compared with NC, the difference was statistically significant, $P < 0.05$). The detection limit of the dual RPA-CRISPR-Cas12 system for *E. coli* was $5.37 \times 10^{1}$ copies/μL. The detection line for *K. pneumoniae* is $5.90 \times 10^{1}$ copies/μL.

### Immunity experiment of dual RPA-CRISPR-Cas12 method

A qualified detection system should have good anti-interference ability. The genomic extracts of interfering bacteria (including *Staphylococcus aureus*, *Pseudomonas aeruginosa*, and *Enterococcus faecalis*) were added to the dual RPA-CRISPR-Cas12 reaction system to simulate the interference of nucleic acid extracts of other bacteria on the reaction in actual detection. As shown in Fig. 7A and B, Fig. 7A is the analysis of the anti-interference ability of *E. coli* RPA-CRISPR-Cas12a. Figure 7B shows the analysis of the anti-interference ability of *K. pneumoniae* RPA-CRISPR-Cas12a. The results showed that (i) there was no statistically significant difference in the fluorescence intensity produced by each interfering bacterial group (including *S. aureus*, *P. aeruginosa,* or *E. faecalis*) and the

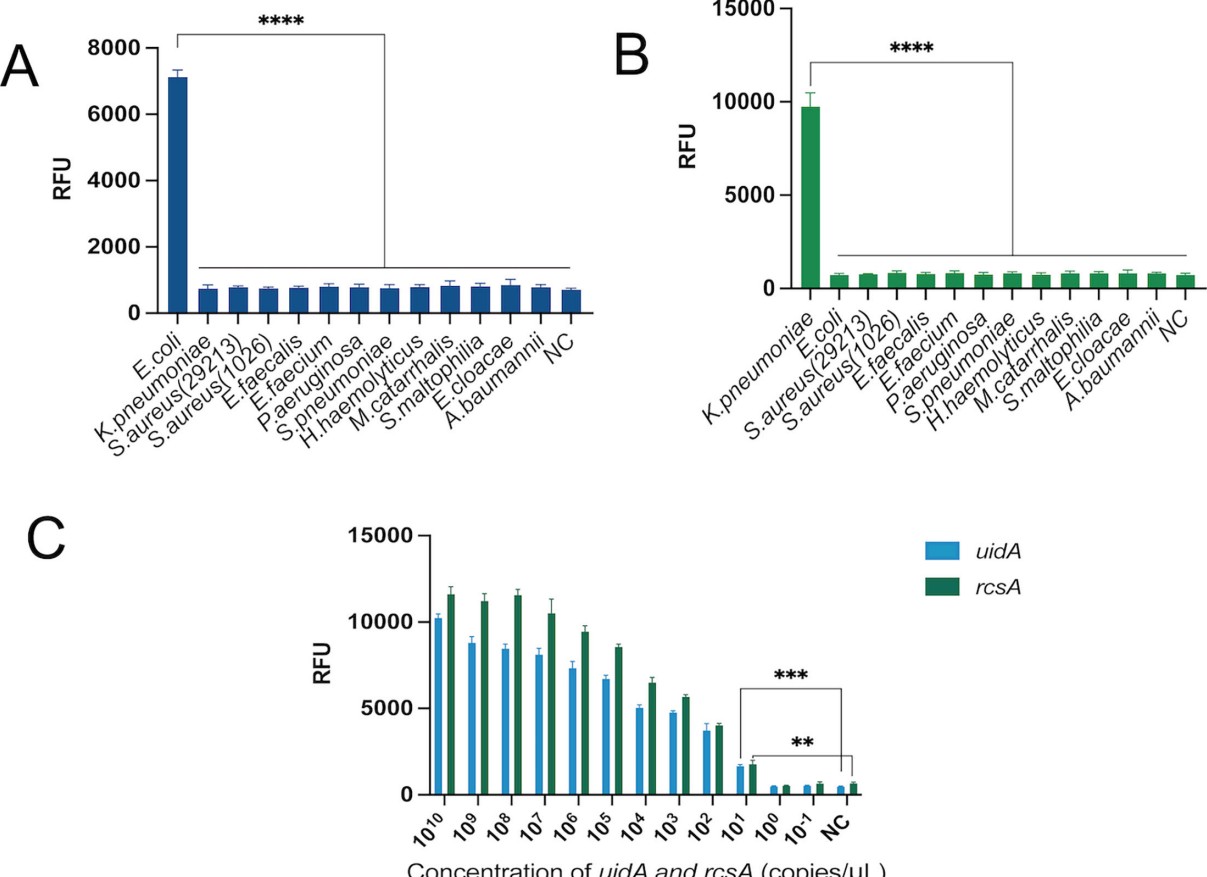

**FIG 6** Sensitivity and specificity of the dual RPA-CRISPR-Cas12 system. (A) *E. coli* specific analysis. (B) *K. pneumoniae* specific analysis. The specificity of the detection system was evaluated using 12 common clinical pathogenic bacteria. The results showed that only *E. coli* and *K. pneumoniae* induced significant fluorescence signals. (C) Sensitivity analysis. The genomic DNA templates of *E. coli* (*uidA* gene) and *K. pneumoniae* (*rcsA* gene) were subjected to a 10-fold gradient dilution (concentration range: $10^{10}$ to $10^{-1}$ copies/μL). Analytical sensitivity refers to the lowest concentration of the analyte at which there is a significant difference between the detected signal and the negative control signal, as obtained through statistical comparison (usually at the significance level $P < 0.05$). It is an important indicator for evaluating the detection capability of analytical methods at low concentration levels, used to determine the lowest concentration level that the method can reliably detect. The experimental results show that this system can stably detect target DNA as low as $10^1$ copies/μL. The final analytical sensitivity was determined to be $5.37 \times 10^1$ copies/μL (uidA) and $5.90 \times 10^1$ copies/μL (*rcsA*). **, $P < 0.01$; ***, $P < 0.001$; ****, $P < 0.0001$.

target bacterial group (only containing *E. coli* + *K. pneumoniae*) ($P > 0.05$); (ii) there was no significant difference in fluorescence intensity between the mixed interfering bacteria group and the negative blank control group ($P > 0.05$). The above results indicate that the dual RPA-CRISPR-Cas12a detection system has good anti-interference ability and can accurately detect the target pathogen (*E. coli* + *K. pneumoniae*) in the presence of potential interfering microorganisms.

## Repeatability test of dual RPA-CRISPR-Cas12 method

We conducted five independent repeated reactions using the standard plasmid with a copy number of $10^4$ to verify the repeatability of the dual RPA-CRISPR-Cas12 detection system. The fluorescence values at the reaction endpoint were taken for the calculation of the mean, standard deviation, and coefficient of variation. The results are shown in Table S3. The relative coefficients of variation of the five independent repeated experiments for *uidA* and *rcsA* were all less than 10%, indicating that the constructed dual RPA-CRISPR-Cas12 detection system has good repeatability.

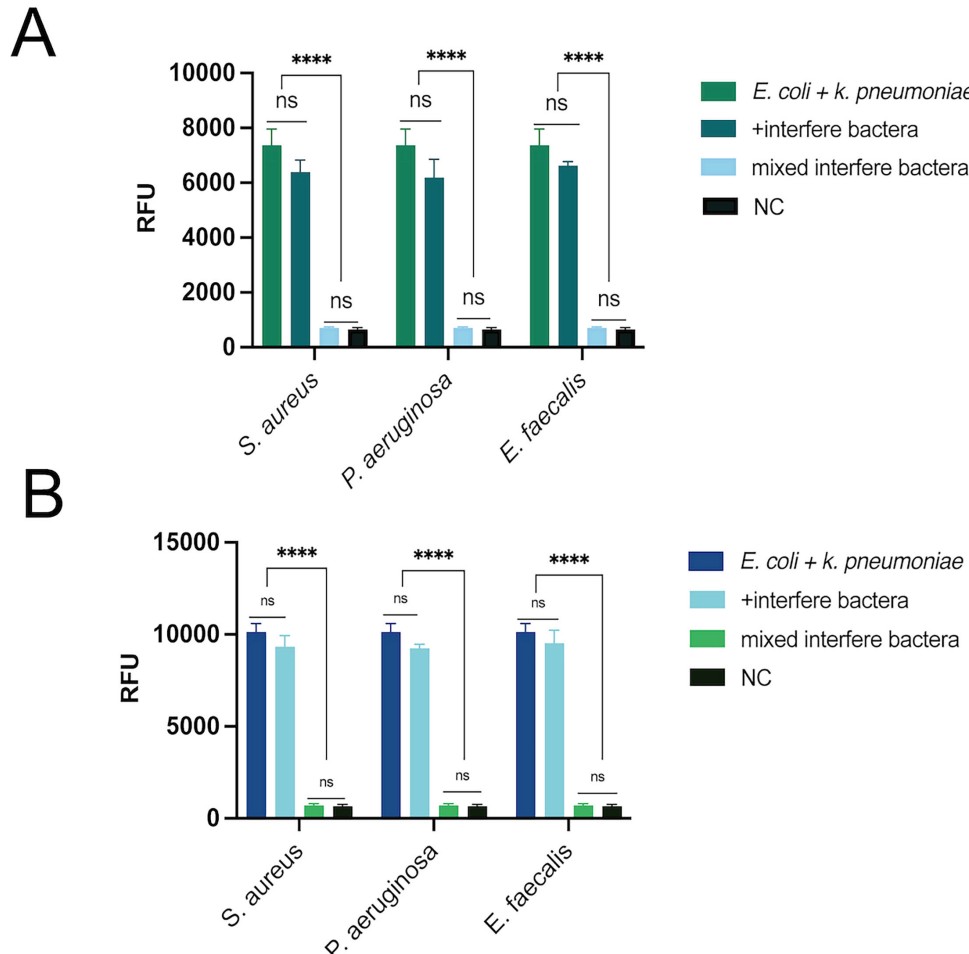

**FIG 7** Anti-interference performance of the dual RPA-CRISPR-Cas12a system. Panel A shows the analysis of the anti-interference ability of *E. coli* RPA-CRISPR-Cas12a. Panel B shows the analysis of the anti-interference ability of *K. pneumoniae* RPA-CRISPR-Cas12a. Experimental grouping to evaluate the anti-interference ability of the detection system, *S. aureus*, *P. aeruginosa*, and *E. faecalis* were selected as interfering bacteria. The following four sets of templates were established for the experiment: (i) target bacterial group: *E. coli* + *K. pneumoniae*; (ii) interfering bacteria group: Target bacteria group (*E. coli* + *K. pneumoniae*) + one interfering bacteria (*S. aureus*, *P. aeruginosa*, or *E. faecalis*); (iii) mixed interfering bacteria group: A mixture of three interfering bacteria (*S. aureus*, *P. aeruginosa*, *E. faecalis*); (iv) negative control group (NC): Nuclease-free water. ****, $P <$ 0.0001; ns, not significant.

## Detection of clinical strains by a dual RPA-CRISPR-Cas12 method

To evaluate the clinical effectiveness of the dual RPA-CRISPR-Cas12 method, we simultaneously conducted PCR and dual RPA-CRISPR-Cas12 combined detections on 30 *E. coli* strains, 30 *K. pneumoniae* strains, and 20 non-*K. pneumoniae and E. coli* strains. The results showed that all 30 strains of *E. coli* and 30 strains of *K. pneumoniae* detected by PCR were positive (Fig. 8A and B), and the detection performance of dual RPA-CRISPR-Cas12 was completely consistent with that of PCR (Fig. 9). Validation data demonstrate that the RPA-CRISPR-Cas12a platform achieves PCR-comparable sensitivity and specificity, establishing a robust methodology for synchronous detection of *E. coli* and *K. pneumoniae*.

## DISCUSSION

Data from the most recent annual report of the China Bacterial Resistance Surveillance Network indicate that *E. coli* and *K. pneumoniae* are the most predominant clinical

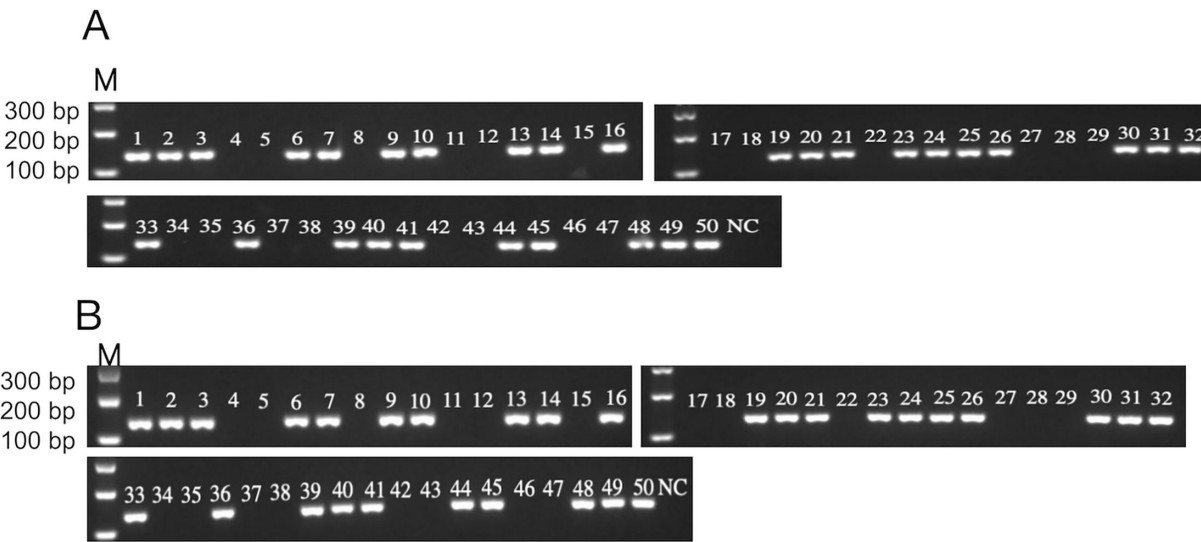

**FIG 8** Detection of clinical isolates of *E. coli* and *K. pneumoniae* by PCR. (A) Agarose gel electrophoresis of PCR amplification products for *E. coli* clinical isolates. A total of 50 clinical samples were tested, including 30 *E. coli*-positive isolates and 20 non-*E. coli* negative controls (e.g., other pathogenic bacteria or non-pathogenic flora, consistent with clinical sample sources). (B) Agarose gel electrophoresis of PCR amplification products for *K. pneumoniae* clinical isolates. A total of 50 clinical samples were tested, including 30 *K. pneumoniae*-positive isolates and 20 non-*K. pneumoniae*-negative controls (same negative control type as in Panel A). M, marker; NC, negative control (nuclease-free water).

isolates (27). Both are prominent pathogens responsible for hospital-acquired infections and can lead to severe complications, including sepsis, urinary tract infections, secondary meningitis, and surgical site infections (28), presenting a substantial challenge to public health. Consequently, the establishment of rapid, sensitive, specific, and reliable detection methods for *E. coli* and *K. pneumoniae* is of great significance in facilitating early and accurate clinical diagnosis and guiding treatment decisions.

Currently, the gold standard for pathogen detection (bacterial culture) typically requires 24–72 h, which fails to meet the demand for rapid diagnosis in critically ill patients. Molecular diagnostic techniques such as PCR can shorten the detection time to 1.5–3.0 h, but they rely on strict temperature cycling procedures, precise instruments, and specialized laboratory settings, which significantly restrict their utilization in point-of-care testing and regions with scarce resources (29). The emergence of isothermal amplification technologies has provided a solution to overcome the drawbacks of PCR. Among these technologies, RPA operates at 37°C–42°C and can effectively amplify target sequences within 5–30 min, significantly simplifying the operational conditions (30). Additionally, the CRISPR-Cas system has emerged as a pivotal technology propelling innovation in molecular diagnostics owing to its high specificity and signal amplification capacity (31). Recently, some methods to combine isothermal amplification technologies with CRISPR-Cas12a have been reported, for example, PCR-CRISPR-Cas12a (2), LAMP-CRISPR-Cas12a (32), Dual RPA-LFD method (33), RCA-CRISPR-Cas12a (34), RAA-TS (35), and HhaI-glyceryl-RPA-Cas12a (36) (Table S4). However, while these methods addressed the long turnaround time of traditional bacterial culture, they still possessed certain limitations. PCR requires complex laboratory equipment and specialized technical expertise (2); LAMP demands a relatively high reaction temperature (65°C) and multiple primer pairs (32); RCA relies on circular DNA templates, whose preparation is complex and may require additional enzymatic reactions or DNA synthesis steps (34). Although lateral flow chromatography (LFD/TS) enables rapid visual readout, result interpretation is susceptible to subjective factors (e.g., variability in test line color intensity among observers), which may limit its use in resource-constrained settings (35). In contrast, RPA offers advantages such as operational simplicity, faster reaction times, and reduced equipment requirements, making it more suitable for rapid on-site detection.

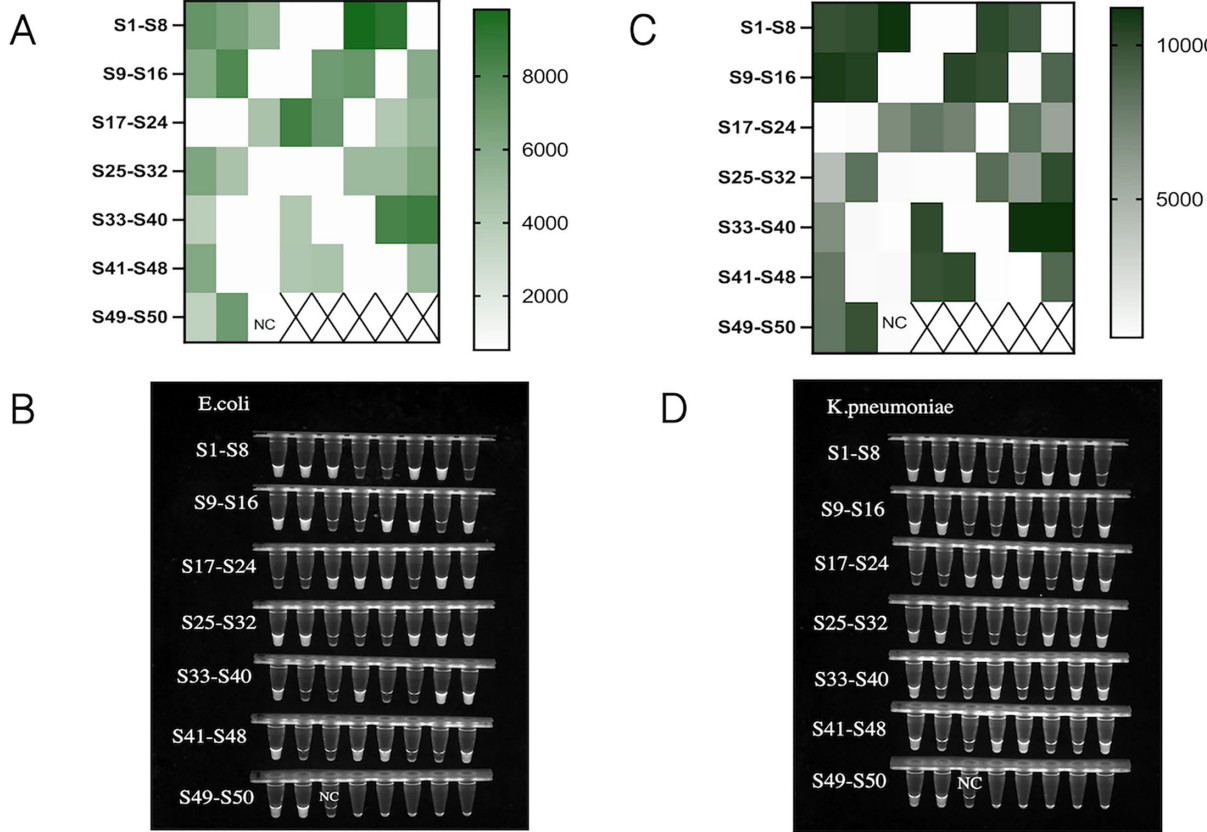

**FIG 9** Validation of clinical samples by dual RPA-CRISPR-Cas12a detection method. (A) Endpoint fluorescence intensity heatmaps of 30 clinical *E. coli* strain samples and 20 negative samples detected based on dual RPA-CRISPR-Cas12a method. (B) Naked-eye visualization of clinical *E. coli* sample test results under ultraviolet lamp excitation. (C) Endpoint fluorescence intensity heatmaps of 30 clinical *K. pneumoniae* strain samples and 20 negative samples were detected based on dual RPA-CRISPR-Cas12a method. (D) Naked-eye visualization observation of clinical sample detection results of *K. pneumoniae* under ultraviolet lamp excitation. NC, negative control.

 With the increasing incidence of mixed infections, single-target nucleic acid testing can hardly meet clinical needs. In this study, we innovatively combined the efficient isothermal amplification of RPA with the high-sensitivity detection of CRISPR-Cas12a, successfully constructing a dual RPA-CRISPR-Cas12a method (where RPA and CRISPR-Cas12a detection are performed in separate tubes). This system enables specific and simultaneous identification of *E. coli* (targeting the *uidA* gene) and *K. pneumoniae* (targeting the *rcsA* gene) in a single detection process. Compared with traditional culture and PCR, this method significantly shortens the total detection time (approximately 70 min) and reduces reliance on precision equipment (endpoint fluorescence can be detected using a real-time PCR instrument or via naked-eye observation under UV lamp excitation). Notably, the optimized microvolume reaction system—requiring only one-fifth of the standard RPA reaction volume—controls the cost per sample to approximately $2, greatly improving its accessibility in resource-limited areas. Under standard laboratory conditions, the analysis sensitivity of this method for *E. coli* and *K. pneumoniae* was $5.37 \times 10^1$ copies/µL and $5.90 \times 10^1$ copies/µL, respectively, with excellent specificity. Validation using clinical samples (showing 100% concordance with PCR results) further confirms its practical application potential.

 However, this study has limitations. First, the current two-step workflow (RPA followed by Cas12a detection) involves manual tube opening and liquid transfer, which introduces the risk of aerosol contamination. Although several "one-pot" strategies have been reported (e.g., light-controlled one-pot [37], glycerol-based one-pot [38], and tube-cap integrated one-pot methods [39]) to enable closed-tube detection and reduce

contamination, most of these approaches are limited to single-target detection. The core challenge lies in the non-specific trans-cleavage activity of Cas12a: once activated, it indiscriminately cleaves all free fluorescent reporter probes in the system, making it impossible to distinguish signals from multiple targets. Thus, developing a "one-pot" method suitable for simultaneous multi-target detection remains a key challenge in this field. Therefore, developing an integrated and closed-tube detection system is our key research direction in the future to fundamentally eliminate pollution risks. Second, nucleic acid extraction in this study was performed using a spin-column kit, which requires 15 min and involves relatively complex steps. While rapid nucleic acid release agents have recently emerged as an alternative for direct nucleic acid extraction from clinical specimens, sputum samples—with their complex components and low nucleic acid content—are prone to false-negative results when processed with these agents.

In conclusion, the focus of our future work will be continuously optimizing the current system and exploring the possibility of integrating multiple RPA with Cas12a multiple detection. Focus on breaking through the technical bottleneck of the "one-pot method" and develop a closed integrated solution that can stably distinguish multiple signals and minimize manual lid opening and liquid transfer to the greatest extent, to reduce operation steps and potential cross-contamination risks. If this goal can be achieved, it will significantly enhance the convenience, biosafety, and multi-detection efficiency of this strategy, making it more suitable for rapid response to infectious diseases and on-site application in resource-limited areas or bedside scenarios.

In conclusion, this study successfully designed and validated a dual RPA-CRISPR-Cas12a method, which enables synchronous and specific detection of *E. coli* (targeting the *uidA* gene) and *K. pneumoniae* (targeting the *rcsA* gene). This system exhibits significant advantages in detection speed, sensitivity, and operational ease, thereby providing an efficient tool for the early and accurate identification of these clinical pathogenic bacteria. Notably, its low requirements for conventional laboratory equipment and controllable testing costs endow it with substantial application potential in resource-limited settings. As a reliable and cost-effective evaluation platform, it can facilitate the early identification of these key pathogens in primary medical institutions, which is crucial for helping mitigate the risk of their spread and transmission.

## ACKNOWLEDGMENTS

This work was supported by the Natural Science Foundation of China under grant no. 82260418 to G.W., the Natural Science Foundation of Guangxi under grant nos. 2024JJH140215 and 2024JJH140265 to L.L., First Batch of High-level Talent Scientific Research Projects of the Affiliated Hospital of Youjiang Medical University for Nationalities under grant no. R202011701 to G.W., the Baise Scientific Research and Technology Development Project under grant nos. 20232031, 20241534, and 20250344 to G.W., and Research Project of Guangxi Zhuang Autonomous Region Disease Prevention and Control Bureau under grant no. GXJKKJ24C010 to G.W.

Conceptualization, L.W., G.W.; formal analysis, X.L., B.R., G.P.; funding acquisition, G.W., L.L.; investigation, S.L., L.L., M.L., G.P.; methodology, L.W.; supervision, L.L., X.L., W.Z.; validation, G.P.; writing—original draft, L.W.; writing—review and editing, L.W., G.W. All authors have read and agreed to the published version of the manuscript.

## AUTHOR AFFILIATIONS

[1]Affiliated Hospital of Youjiang Medical University for Nationalities, Baise, Guangxi, China
[2]Baise Key Laboratory of In Vitro Diagnostic Technology, Baise, Guangxi, China
[3]Guangxi Key Laboratory of Artificial Intelligence for Genetic Diseases of Long- dwelling Nationalities, Baise, Guangxi, China
[4]Guangxi Engineering Research Center for Precise Genetic Testing of Long-dwelling Nationalities, Baise, Guangxi, China

[5]Key Laboratory of Research on Clinical Molecular Diagnosis for High Incidence Diseases in Western Guangxi of Guangxi Higher Education Institutions, Baise, Guangxi, China
[6]Baise People's Hospital, Baise, Guangxi, China
[7]Yaneng BIOscience (Shenzhen) Corporation, Shenzhen, Guangdong, China
[8]Clinical Genome Center, Guangxi KingMed Diagnostics, Nanning, Guangxi, China

## AUTHOR ORCIDs

Guijiang Wei http://orcid.org/0009-0004-5473-009X
Xuebin Li http://orcid.org/0009-0004-9684-2428
Lina Liang http://orcid.org/0009-0003-6211-7204

## AUTHOR CONTRIBUTIONS

Lijian Wei, Conceptualization, Methodology, Writing – original draft, Writing – review and editing | Guangfu Pang, Formal analysis, Investigation, Supervision | Shihua Luo, Investigation | Weijie Zhou, Supervision | Baoyan Ren, Formal analysis | Miao Li, Investigation | Guijiang Wei, Conceptualization, Funding acquisition, Writing – review and editing | Xuebin Li, Formal analysis, Supervision | Lina Liang, Funding acquisition, Investigation, Supervision

## DATA AVAILABILITY

This study was carried out using publicly available data from NCBI at http://www.ncbi.nlm.nih.gov/ with *K. pneumoniae* rcsA gene (GenBank: AY059955.1) and the *E. coli* uidA gene (GenBank: KT311783.1).

## ETHICS APPROVAL

The clinical specimens were sourced from the Microbiology Laboratory of the Laboratory Department of the Affiliated Hospital of Youjiang Medical University for Nationalities. All clinical strains were identified using the VITEK2 compact microbial identification system (Lyon, France). All bacterial cultures were completed in the Microbiology Laboratory of the Affiliated Hospital of Youjiang Medical University for Nationalities and were approved by the Ethics Committee of the Affiliated Hospital of Youjiang Medical University for Nationalities (2025021901).

## ADDITIONAL FILES

The following material is available online.

### Supplemental Material

**Supplemental material (Spectrum03598-25-s0001.docx).** Tables S1 to S5.

### Open Peer Review

**PEER REVIEW HISTORY (review-history.pdf).** An accounting of the reviewer comments and feedback.

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
