## [Reviewer comments · Microbiology Spectrum]

Microbiology Spectrum

Rapid and Simultaneous Detection of *Escherichia coli* and *Klebsiella pneumoniae*: A Novel Dual Recombinase Polymerase Amplification-Clustered Regularly Interspaced Short Palindromic Repeats/Cas12a Method

Lijian Wei, Guangfu Pang, Shihua Luo, Weijie Zhou, Baoyan Ren, Miao Li, Guijiang Wei, Xuebin Li, and Lina Liang

Corresponding Author(s): Lina Liang, Affiliated Hospital of Youjiang Medical University for Nationalities, Baise, Guangxi, 533000, China.

Review Timeline:

Submission Date:	November 7, 2025
Editorial Decision:	January 19, 2026
Revision Received:	January 27, 2026
Accepted:	February 22, 2026

Editor: Vittal Ponraj

Reviewer(s): The reviewers have opted to remain anonymous.

Transaction Report:

DOI: <https://doi.org/10.1128/spectrum.03598-25>

Re: Spectrum03598-25 (Rapid and Simultaneous Detection of Escherichia coli and Klebsiella pneumoniae: A Novel Dual Recombinase Polymerase Amplification-Clustered Regularly Interspaced Short Palindromic Repeats/Cas12a Method)

Dear Mrs. Lina Liang:

Thank you for the privilege of reviewing your work. Below you will find my comments, instructions from the Spectrum editorial office, and the reviewer comments.

Revision Guidelines

Sincerely,

Vittal Prakash Ponraj Ph.D
Editor
Microbiology Spectrum

Reviewer #1 (Comments for the Author):

The manuscript by Wei et al. describes a dual RPA-CRISPR/Cas12a fluorescence assay for detecting Escherichia coli and Klebsiella pneumoniae, two important hospital-associated pathogens. The assay targets uidA for E. coli and rcsA for K. pneumoniae and is reported to achieve low limits of detection with a relatively short turnaround time. Overall, the strategy is promising and could provide a useful alternative approach for rapid diagnosis of E. coli and K. pneumoniae infections.

Overall, the study is well-designed, and the workflow is generally easy to follow, though the clarity of several experimental details need to be improved. The results are interpreted appropriately and limitations of the study are clearly discussed. I have some concerns, which I hope to help the authors strengthen their manuscript.

Major concerns

1. There seems to be inconsistency about CRISPR/Cas12a component concentrations between Results and Methods. In Results, the optimized Cas12a/crRNA/reporters are in the 200-600 nM range (e.g., Cas12a 300 nM for uidA; 200 nM for rcsA; reporters 600/500 nM). But in Methods (CRISPR/Cas12a cleavage assay), the reaction is described as 50 nM LbCas12a and 50 nM crRNA, which conflicts with the optimization results.
2. The manuscript alternates between "clinical samples/specimens" and "clinical strains/isolates." The Methods indicate that specimens were collected and confirmed by culture, but the validation dataset (30 *E. coli*, 30 *K. pneumoniae*, 20 negatives) reads more like testing cultured isolates rather than primary specimens (e.g., blood/urine/sputum). Please clarify exactly what was tested.
3. As the workflow includes tube opening and transfer between amplification and detection, the risk of carryover contamination is non-trivial. Although the Discussion acknowledges this, the Methods do not describe concrete mitigation steps (e.g., physical separation of pre-/post-amplification areas, closed-tube strategies, carryover prevention). In addition, there is no extraction/amplification internal control to flag false negatives due to inhibition or extraction failure, which is an important consideration if the method is intended for clinical use.
4. The authors define the "LOD" as "lowest concentration with significant difference vs negative control ($P < 0.05$).". Based on my understanding, this is not the typical approach for diagnostic assays, where LOD is often defined by detection probability (e.g., {greater than or equal to}95% positive across replicates) and may be supported by probit analysis.
5. I am a little bit confused regarding the Dual RPA reaction setup. In the Methods section line 382-390, the Dual-RPA section describes adding multiple microliter volumes into the lyophilized RPA pellet but later states a total reaction volume of 10 μL (8.7 μL premix + template + MgOAc). As written, the step-by-step setup does not clearly add up to the final volume. Please rewrite this section to explicitly list per-reaction component volumes and final concentrations.

Minor comments:

1. Please justify why uidA and rcsA were selected as targets and whether closely related organisms could cause cross-reactivity.
2. The usage of KP to represent *Klebsiella pneumoniae* is not scientifically right. Please use consistent species formatting (*E. coli*, *K. pneumoniae*) and consistent gene formatting (uidA, rcsA). Several sections mix KP/*E. coli*, "*E. coli*," "*the E. coli*," etc. All species and gene names need to be italicized.
3. There are many typos throughout the manuscript. For example: "targeting the the" in line 139, "trans - cleavage" in line 79.
4. In line 268, it is not clear to me what "CHINET 2024" means. Please provide a proper citation.
5. The manuscript describes PAM as 5'-TTTN-3' and refers to "cis-cleavage property precisely cuts the PAM site." This should be revised for mechanistic accuracy. Cas12a cleaves target DNA at defined positions relative to PAM rather than cutting the PAM itself.
6. Figure 8: label the marker size.

Reviewer #2 (Comments for the Author):

The manuscript entitled "Rapid and Simultaneous Detection of *Escherichia coli* and *Klebsiella pneumoniae*: A Novel Dual Recombinase Polymerase Amplification-Clustered Regularly Interspaced Short Palindromic Repeats/Cas12a Method" describes a rapid and potentially cost-effective approach for detecting *E. coli* and *K. pneumoniae* in clinical applications. Overall, the study presents the development and optimization of a dual RPA-CRISPR/Cas12a assay and demonstrates promising sensitivity and specificity. However, several issues should be addressed:

Major Comments

1. The authors state that the dual RPA-CRISPR/Cas12a assay can specifically and simultaneously identify *E. coli* (targeting the uidA gene) and KP (targeting the rcsA gene) in a single detection process. However, based on fluorescence readouts, detection of *E. coli* and KP appears to be performed in separate tubes. Thus, the system is dual in terms of gene amplification rather than signal detection. This distinction should be clearly clarified in the text (Only mentioned in Discussion).
2. In lines 168-171, the differences between crRNA1 and crRNA2 are not clearly described. Please explain their respective roles and design rationale in the main text.
3. In the section "Optimization of the Dual RPA-CRISPR/Cas12a Method" (lines 176-192), it is unclear what criteria were used to

compare band quality across different conditions. Please specify how amplification performance was evaluated (e.g., band intensity, specificity, absence of smearing).

4. In lines 201-215, optimization was performed by varying one factor at a time. Since this approach does not necessarily identify the optimal overall condition, please provide a direct comparison between the baseline (unchanged) condition and the conditions obtained after optimizing each individual factor.

5. For the clinical strain validation, the manuscript does not clearly describe how clinical samples were processed prior to the RPA-CRISPR/Cas12a assay. Please clarify the sample preparation workflow, including any extraction or pretreatment steps.

Minor Comments

1. Some references appear to be inappropriate or not well matched to the statements they support; please review and revise accordingly.

2. There are formatting inconsistencies in Figures 2 and 3 (e.g., "300bp" vs "300 bp"), as well as inconsistent font sizes in Figures 4 and 5. Overall, figures should be better standardized and organized.

3. The term "screening" used in Figure 2 may be misleading given the limited number of conditions tested. Terms such as "comparison" or "evaluation" would be more appropriate.

4. Please ensure consistent formatting of bacteria names (italicized) throughout the manuscript and figure legends.

5. Minor grammatical and typographical errors are present and should be corrected throughout the text.

Reviewer #1 (Comments for the Author):

The manuscript by Wei et al. describes a dual RPA-CRISPR/Cas12a fluorescence assay for detecting *Escherichia coli* and *Klebsiella pneumoniae*, two important hospital-associated pathogens. The assay targets *uidA* for *E. coli* and *rcaA* for *K. pneumoniae* and is reported to achieve low limits of detection with a relatively short turnaround time. Overall, the strategy is promising and could provide a useful alternative approach for rapid diagnosis of *E. coli* and *K. pneumoniae* infections.

Overall, the study is well-designed, and the workflow is generally easy to follow, though the clarity of several experimental details needs to be improved. The results are interpreted appropriately, and limitations of the study are clearly discussed. I have some concerns, which I hope to help the authors strengthen their manuscript.

Major concerns

1. There seems to be an inconsistency about CRISPR/Cas12a component concentrations between Results and Methods. In Results, the optimized Cas12a/crRNA/reporters are in the 200-600 nM range (e.g., Cas12a 300 nM for *uidA*; 200 nM for *rcaA*; reporters 600/500 nM). But in Methods (CRISPR/Cas12a cleavage assay), the reaction is described as 50 nM LbCas12a and 50 nM crRNA, which conflicts with the optimization results.

Dear reviewer:

Thank you very much for your meticulous review of our manuscript and for your valuable comments on the consistency of the CRISPR/Cas12a component concentration in the results and methods section. The issue you pointed out is indeed very important and provides a good direction for us to further clarify the information in the text. Regarding the concentration difference you mentioned, the following explanation is provided here:

The CRISPR/Cas12a reaction system described in the method section (containing 50 nM LbCas12a and 50 nM crRNA) was the initial concentration we adopted when initially establishing the reaction conditions, which was used to verify the feasibility of the detection system in principle. After confirming the feasibility of the system, in order to further enhance the detection sensitivity and signal strength, we carried out systematic optimization of the concentration of each component.

The optimization results show that the optimal response concentrations vary for different target genes (*rcaA* and *uidA*): For *K. pneumoniae* (*rcaA*), the optimal reaction conditions are: 200 nM LbCas12a, 200 nM crRNA, and 500 nM fluorescent reporter molecule; For *E. coli* (*uidA*), the optimal conditions are: 300 nM LbCas12a, 300 nM crRNA, and 600 nM fluorescent reporter molecule.

This optimization process and the ultimately adopted optimal concentration have been reported in the results section, but the methods section has not been updated simultaneously. We sincerely apologize for any confusion this may cause you. To maintain consistency throughout the text and avoid misunderstandings, we will make

the following updates to the methods section in the revised draft (marked in red):
Description of the revised method:(209-216)

K. pneumoniae (rcsA) testing system: The 20 μ L reaction system contains 2 μ L of 10 \times NEBuffer r2.1, 2 μ L of amplification products, 500 nM fluorescent reporter molecule (ssDNA), 200 nM LbCas12a, and 200 nM crRNA, which is replenished to 20 μ L with nuclease-free water.

E. coli (uidA) detection system: The 20 μ L reaction system contains 2 μ L of 10 \times NEBuffer r2.1, 2 μ L of amplification products, 600 nM fluorescent reporter molecule (ssDNA), 300 nM LbCas12a, and 300 nM crRNA, which is replenished to 20 μ L with nuclease-free water.

We believe that the above revisions can more accurately and clearly reflect the final conditions used in the experiment. Thank you again for your rigorous review and constructive suggestions, which will be of great help in improving the quality of the manuscript. If you have any further questions, we would be more than happy to provide additional explanations.

2.The manuscript alternates between "clinical samples/specimens" and "clinical strains/isolates." The Methods indicate that specimens were collected and confirmed by culture, but the validation dataset (30 *E. coli*, 30 *K. pneumoniae*, 20 negatives) reads more like testing cultured isolates rather than primary specimens (e.g., blood/urine/sputum). Please clarify exactly what was tested.

Dear reviewer:

Thank you very much for raising this important and detailed question. The inconsistency of the terms you pointed out may indeed cause misunderstandings. We sincerely apologize for this and are more than willing to clarify it. Your understanding is very correct. The workflow of this study does indeed consist of two stages:

(i) Clinical sample collection and strain isolation: We collected original clinical specimens (including blood, urine, and sputum) from the Affiliated Hospital of Youjiang Medical University for Nationalities. All these specimens have undergone standard clinical microbiological cultures to confirm the presence of pathogens (*E. coli*, *K. pneumoniae*) and to isolate purified clinical strains. Subsequently, DNA from these clinical strains was extracted to detect RPA-CRISPR-Cas12a.

(ii) Establishment and validation of the RPA-CRISPR-Cas12a detection method: At this stage, to evaluate the specificity, sensitivity and accuracy of our newly established detection method under controllable conditions and avoid the possible inhibition or interference caused by complex matrices (such as blood components, mucus, etc.) in the original specimens, we used genomic DNA extracted from the above-mentioned clinical isolates as the template for the verification experiment. Therefore, the 30 strains of *E. coli*, 30 strains of *K. pneumoniae*, and 20 negative controls that you saw in the "Validation Dataset" section of the manuscript were indeed tested for the genomic DNA of these clinical isolates, rather than direct lysis detection of the original specimens.

We agree that the description of this in the method section is not clear enough.

In response to the issue of term confusion you pointed out, we will make the following modifications in the revised draft to maintain clarity and consistency: In the "Clinical Sample Testing" section, it is clearly described as: "We collected original clinical specimens (including blood, urine, sputum, etc.) from the Affiliated Hospital of Youjiang Medical University for Nationalities." All these specimens have undergone standard clinical microbiological cultures to confirm the presence of pathogens (such as *E. coli*, *K. pneumoniae*, etc.) and to isolate purified clinical strains. Genomic DNA was extracted from the purified clinical strains for the subsequent detection of RPA-CRISPR-Cas12a. (292-297)

3. As the workflow includes tube opening and transfer between amplification and detection, the risk of carryover contamination is non-trivial. Although the Discussion acknowledges this, the Methods do not describe concrete mitigation steps (e.g., physical separation of pre-/post-amplification areas, closed-tube strategies, carryover prevention). In addition, there is no internal extraction/amplification internal control to flag false negatives due to inhibition or extraction failure, which is an important consideration if the method is intended for clinical use.

Dear reviewer:

Thank you very much for putting forward these crucial opinions. Your suggestions on pollution risk control and internal quality control are of great guiding significance for us to enhance the rigor and clinical applicability of our methods. We are deeply grateful for this and hope to clarify and supplement your concerns as follows:

Regarding the prevention and control measures for pollution risks: The pollution risks you pointed out regarding the open operation process are completely correct. During the experiment, we took a series of strict preventive measures to minimize the risk of carrying contamination, but we are sorry that they were not detailed in the methods section of the manuscript. We will immediately add the following specific operational details in the methods section:(142-153)

(i) Zoning operation: All experimental steps are carried out in physically separated Spaces. Reagent preparation and sample processing are carried out in the clean preparation area, PRA amplification is conducted in the amplification area, and the opening of the amplification product caps and CRISPR detection are completed in the independent product analysis area, following a one-way workflow.

(ii) Anti-contamination operation: Use pipette tips with filter cores throughout the process. Laboratory personnel should wear gloves and change them frequently. After each experimental batch, thoroughly clean the working area with nucleic acid remover.

(iii) Negative control setup: We set up a template-free control (NTC) in each experimental batch to monitor whether there is any contamination of amplification products from reagents or the environment throughout the process. All NTC results were negative, confirming that carried contamination was effectively controlled under our operating conditions.

(iv) Future Direction: We will add explanations in the discussion section, pointing

out that developing an integrated, closed-tube detection system is our key research direction in the future to fundamentally eliminate pollution risks. (537-539)

Internal quality control regarding monitoring false negatives:

We fully understand and agree with your core concerns. Our current negative control (NTC) is mainly used to monitor "contamination", but as you pointed out, there is indeed a lack of an "internal control" integrated into the reaction system to monitor false negatives caused by "reaction failure". We have noticed that in the early development stage of the current CRISPR diagnostic technology, most of the similar studies focused on principle verification. For example, "An Extraction-Free One-Pot Assay for Rapid Detection of *Klebsiella pneumoniae* by Combining RPA and" published by Fu et al "CRISPR/Cas12a" and "A CRISPR/Cas12a-Assisted SERS Nanosensor for Highly Sensitive Detection of HPV DNA" published by Ye et al. have not yet integrated internal quality control of samples. "Internal control" to monitor false negatives caused by "reaction failure" is indeed a key focus of our future research on "internal quality control", thereby ensuring the reliability of negative results.

Thank you again for your profound review comments. By supplementing the above details and future, we hope to make the description of the manuscript complete and more rigorous, and clearly demonstrate the feasible path for the development of this method towards clinical application. We hope these clarification and revision plans can address your concerns.

4. The authors define the "LOD" as "lowest concentration with significant difference vs negative control ($P < 0.05$).". Based on my understanding, this is not the typical approach for diagnostic assays, where LOD is often defined by detection probability (e.g., {greater than or equal to}95% positive across replicates) and may be supported by probit analysis.

Dear reviewer:

Thank you for this very important and professional suggestion you put forward. We completely agree with you. Regarding the standard definition of LOD: The LOD you pointed out is determined based on the detection probability (for example, $\geq 95\%$ positive detection rate) and through models such as probit or logistic regression. This method can describe the detection limit of the analyte more accurately and provide a clear detection confidence level.

Regarding the definition of "LOD" in this article: In this article, "LOD" is defined as "the lowest concentration that is significantly different from the negative control ($P < 0.05$)". This essentially determines the lower bound of the analytical sensitivity or detection limit of the detection method. It answers the question of "At what concentration point does the signal start to clearly distinguish from background noise?" Although this definition was common in early methodological research and can provide valuable information, it cannot be directly equated with a concentration that can be stably detected with a high probability (such as 95%) (i.e., the traditional LOD).

To ensure the accuracy of the term usage and enhance the rigor of the method, we have made the following modifications: In line 852-857, LOD is changed to

"Analytical sensitivity", and it is defined as: "The lowest analyte concentration at which the detection signal is significantly different from the negative control signal, obtained through statistical comparison." Through the above modifications, not only can the actual work of this research be accurately reflected, but also your valuable professional opinions have been fully adopted, making the methodological description more standardized and rigorous.

5. I am a little bit confused regarding the Dual RPA reaction setup. In the Methods section line 382-390, the Dual-RPA section describes adding multiple microliter volumes into the lyophilized RPA pellet but later states a total reaction volume of 10 μL (8.7 μL premix + template + MgOAc). As written, the step-by-step setup does not clearly add up to the final volume. Please rewrite this section to explicitly list per-reaction component volumes and final concentrations.

Dear Reviewer:

Thank you very much for pointing out the unclear points in the methodological description and guiding us to conduct a thorough reflection on this. After carefully checking the experimental records, we realized that the confusion in the original description originated from the failure to clearly explain the unique strategy we adopted to save costs, which was to "first prepare 5 reaction parts of concentrated premix and then evenly divide it", and omitted the description of the four-primer system. We sincerely apologize for this. The following is the "Dual-RPA Reaction System" section rewritten based on our actual and precise experimental procedures. This description clearly defines the volume and operation of each step.

2.4 Dual-RPA Amplification System:(183-206)

The Dual-RPA reaction in this study aims to simultaneously detect two targets (the *rcsA* gene of *K. pneumoniae* and the *uidA* gene of *E. coli*), a total of four primers targeting both targets are included in a single 10 μL reaction system. To save reagent costs, we adopt a strategy of first preparing concentrated mother liquor with multiple reaction parts and then evenly dividing it. The specific steps are as follows:

(i) Preparation and aliquoting of 5 reaction parts of premixed mother liquor: Take a complete TwistAmp™ Basic freeze-dried RPA enzyme sphere.

Use a micropipette to add the following components to the enzyme ball to prepare a premixed stock solution sufficient for 5 reactions: 10 \times TwistAmp™ Basic buffer: 29.5 μL , nuclease-free water: 4.4 μL , *K. pneumoniae* forward primer (10 μM): 2.4 μL , *K. pneumoniae* reverse primer (10 μM): 2.4 μL , *E. coli* forward primer (10 μM): 2.4 μL , *E. coli* reverse primer (10 μM): 2.4 μL , total volume: 43.5 μL . Repeatedly pipette or briefly vortex with a pipette to ensure that the freeze-dried enzyme spheres are completely dissolved and well mixed. The above 43.5 μL premixed mother liquor was evenly aliquoted into five independent reaction tubes, with each tube receiving 8.7 μL .

(ii) Completion of a single reaction (10 μL system): Add the following in sequence to the reaction tube that already contains 8.7 μL of premix:

K. pneumoniae DNA template: 0.4 μL , *E. coli* DNA template: 0.4 μL . (Note: In the negative control, both templates should be replaced with equal volumes of nuclease-free water.) Gently pipette and mix well. Finally, add 0.5 μL of 280 mM magnesium acetate (MgOAc) solution to initiate the reaction and immediately centrifuge briefly. Reaction procedure: Immediately place the reaction tube in a constant temperature metal bath at 37°C and incubate for 25 minutes. The Dual-RPA

reaction system is shown in Supplementary Materials Table S5.

constituents	volume (μL)	concentration
Premix (containing enzymes, buffer solution, enzyme-free water, primers)	8.7	-
K. pneumoniae (rcsA) DNA template	0.4	-
E. coli (uidA) DNA template	0.4	-
MgOAc	0.5	14mM
Total volume of a single reaction	10	-
Premix		
Basic buffer	5.9	-
K. pneumoniae F/R primers	0.48/ 0.48	480nM
E. coli F/R primers	0.48 /0.48	480nM
enzyme-free water	0.88	-
Total volume of the single-component premix	8.7	-

Minor comments:

1. Please justify why *uidA* and *rcsA* were selected as targets and whether closely related organisms could cause cross-reactivity.

Dear reviewer:

Thank you for raising this profound question. We chose *uidA* and *rcsA* as detection targets based on their high species specificity. More importantly, through a series of rigorous experiments, including broad-spectrum specificity verification and anti-interference tests, we have fully demonstrated that the constructed dual RPA-CRISPR/Cas12a detection system does not have cross-reactions caused by related species or other common pathogens.

(i) *uidA* gene targeting *E. coli*:

The *uidA* gene (encoding β -glucuronidase) is recognized as a highly specific and reliable genetic marker for *Escherichia coli* detection. As published by Suzuki et al., "Simultaneous detection of various pathogenic bacteria".

The study "Escherichia coli in water by sequencing multiplex PCR amplicons" clearly pointed out that *uidA* was selected as the detection target of universal *Escherichia coli*, listed alongside other pathogenic genes (such as *stx1*, *stx2*, etc.), proving its stability in complex detection systems.

Furthermore, Walker et al. published "A highly specific *Escherichia coli* qPCR and its comparison with existing methods for environmental the "waters" study also clearly indicates that the qPCR detection method based on *uidA* shows high inclusiveness for *Escherichia coli* (up to 98.9%) and has good quantitative ability (amplification efficiency 89.8%, $R^2 = 0.998$), making it suitable for the accurate quantification of *E. coli* in environmental water bodies. This study also compared multiple methods and further confirmed the effectiveness of *uidA* as a biomarker for *E.*

coli detection.

(ii) *rcsA* gene against *K. pneumoniae*:

The selection of the *rcsA* gene is mainly based on its high species specificity for *K. pneumoniae*. "Survey and rapid detection of *Klebsiella pneumoniae* in clinical samples targeting the *rcsA* gene in Beijing" published by Dong et al. The "China" study specifically developed a LAMP detection method for the *rcsA* gene and confirmed that the detection limit of this method was as low as 0.115 pg/ μ L, and the test results for 30 non-*K. pneumoniae* strains were all negative, demonstrating 100% specificity.

In conclusion, our selection of *uidA* and *rcsA* as detection targets is based on the abundant experimental evidence in the existing literature, and comprehensively considers the verification results of their species specificity and detection sensitivity. We believe that the selection of these two targets is reasonable and targeted and can meet the requirements of this study for accurate and specific pathogen detection.

(iii) Systematic experimental verification for cross-reactivity

We attach great importance to the potential cross-reaction risks brought by closely related species and have conducted systematic and multi-level verifications on this. The verification results are clearly presented in Figures 6 and 7 of the manuscript:

Specificity verification (corresponding to Figures 6A and 6B): We tested the detection system using genomic DNA from a series of common clinical pathogenic bacteria (a total of 12 species). The experimental results clearly show that only when the target bacteria (*E. coli* or *K. pneumoniae*) are present will a strong specific fluorescence signal be produced, while all other tested strains (including possible closely related species) do not produce any significant signals. This directly proves that our detection system can accurately distinguish the target bacteria from their closely related species, demonstrating outstanding specificity.

Anti-interference capability verification (corresponding to Figures 7A and 7B): To simulate more complex actual sample environments (such as mixed infections), we specially designed anti-interference experiments. In the presence of the target bacteria (*E. coli* + *K. pneumoniae*), we additionally added a variety of common interfering bacteria (*S. aureus*, *P. aeruginosa*, *E. faecalis*). The experimental results show that the presence of these interfering bacteria does not significantly inhibit or enhance the detection signal of the target bacteria ($P > 0.05$). The group containing only mixed interfering bacteria did not produce any specific signals, just like the negative control group. This strongly confirms the stability and reliability of the detection system in the face of complex microbial backgrounds, eliminating the possibility of non-specific cross-reactions.

In conclusion, *uidA* and *rcsA* are carefully selected ideal targets based on their species specificity. Through broad-spectrum specific tests covering a variety of common clinical pathogens (including related species) and anti-interference experiments simulating complex samples, we have provided strong experimental evidence, confirming that the dual RPA-CRISPR/Cas12a detection method we developed has high specificity and can effectively distinguish target bacteria from related species. There is no cross-reaction. The experimental verification of these systems ensures the accuracy and reliability of this method in clinical practical applications.

2.The usage of KP to represent *Klebsiella pneumoniae* is not scientifically right. Please use consistent species formatting (*E. coli*, *K. pneumoniae*) and consistent gene formatting (*uidA*, *rcsA*). Several sections mix KP/*E. coli*, "*E. coli*," "the*E. coli*," etc. All species and gene names need to be italicized.

Dear Reviewer:

Thank you very much for your careful review and valuable suggestions. The issues you pointed out regarding the inconsistent format of species and gene names and the improper use of italics are completely correct. This is an oversight in our process of writing and revising the manuscript, for which we sincerely apologize. We have conducted a systematic review and revision of the entire text based on your suggestions to ensure the rigor of terms used and the standardization of the format. The specific modifications are as follows:

Species name: All references to *Klebsiella pneumoniae* in the entire text have been uniformly revised to *K. pneumoniae* (the full name was used when it first appeared). All references to *Escherichia coli* in the entire text have been uniformly revised to *E. coli* (the full name is used when it first appears). The format of all species names has been checked and corrected, and it has been ensured that they are all in italics in the main text. Gene names: The gene names involved in the text (such as *uidA*, *rcsA*, etc.) have been uniformly set in italics.

We have updated the above modifications to the manuscript. Thank you again for your rigorous academic attitude and professional guidance, which have greatly enhanced the scientific rigor of our manuscripts. We will check the details of the manuscript more carefully in the future.

If you find any other problems while reviewing the revised draft, please feel free to raise them at any time.

3.There are many typos throughout the manuscript. For example: "targeting the the" in line 139, "trans - cleavage" in line 79.

Dear Reviewer:

We sincerely thank the reviewer for identifying the typographical errors in our manuscript. We apologize for these oversights.

In response, we have performed a comprehensive spell-check and manual proofreading of the entire text. The specific errors mentioned ("targeting the the" and "trans - cleavage") have been corrected, and we have rectified similar typos and formatting inconsistencies throughout the manuscript.

We believe these corrections have enhanced the readability and accuracy of our work. Thank you again for the valuable feedback.

4. In line 268, it is not clear to me what "CHINET 2024" means. Please provide a proper citation.

Dear Reviewer:

Thank you for raising this point that needs clarification. In line 268, "CHINET 2024" refers to the latest annual monitoring report from the CHINET (China Bacterial Resistance Surveillance Network) project, which is a national, multi-center bacterial resistance surveillance system. To avoid ambiguity and provide standardized references, we have modified the text accordingly.

In the revised draft, line 268 is now changed to: "Data from the most recent annual report of the China Bacterial Resistance Surveillance Network (CHINET) indicate that *E. coli* and *K. pneumoniae* are the most predominant clinical isolates."

We have also added the corresponding citation entries (numbered [27]) to the reference list. Thank you for prompting us to enhance the clarity and academic rigor of this citation.

5. The manuscript describes PAM as 5'-TTTN-3' and refers to "cis-cleavage property precisely cuts the PAM site." Th should be revised for mechanistic accuracy. Cas12a cleaves target DNA at defined positions relative to PAM rather than cutting the PAM itself.

Dear Reviewer:

Thank you very much for your meticulous review and valuable professional advice. The issue you pointed out regarding the inaccurate description of the Cas12a cutting mechanism is of great significance. We fully agree and apologize for this. The original text's expression is indeed prone to misunderstanding and fails to accurately reflect the positional relationship of its cutting site relative to PAM.

Based on your suggestions, we have made the following modifications to the relevant parts of the manuscript:

The original manuscript statement (inaccurate):

"... Precisely cut PAM sites by taking advantage of its cis-cutting characteristics."

Revised expression:

"... **By leveraging its cis-cleavage activity, target DNA strands are cleaved at specific sites downstream of the PAM sequence. (326-327)**

Once again, we sincerely thank you for helping us enhance the scientific rigor of our thesis. We have corrected this part and other related descriptions in this round of revision. We look forward to your further guidance.

6. Figure 8: label the marker size.

Dear Reviewer:

Thank you, reviewer, for pointing out this issue. You're completely right. Based on your suggestions, we have clearly marked the DNA fragment size (bp) corresponding to the key bands beside the Marker Lane in the modified Figure 8. This modification makes the result clearer and more readable.

Reviewer #2 (Comments for the Author):

The manuscript entitled "Rapid and Simultaneous Detection of Escherichia coli and Klebsiella pneumoniae: A Novel Dual Recombinase Polymerase Amplification-Clustered Regularly Interspaced Short Palindromic Repeats/Cas12a Method" describes a rapid and potentially cost-effective approach for detecting E. coli and K. pneumoniae in clinical applications. Overall, the study presents the development and optimization of a dual RPA-CRISPR/Cas12a assay and demonstrates promising sensitivity and specificity. However, several issues should be addressed:

Major Comments

1. The authors state that the dual RPA-CRISPR/Cas12a assay can specifically and simultaneously identify *E. coli* (targeting the *uidA* gene) and KP (targeting the *rcsA* gene) in a single detection process. However, based on fluorescence readouts, detection of *E. coli* and KP appears to be performed in separate tubes. Thus, the system is dual in terms of gene amplification rather than signal detection. This distinction should be clearly clarified in the text (Only mentioned in the discussion).

Dear Reviewer:

Thank you very much for your profound and professional opinion. You have accurately pointed out the possible ambiguity in the description of "double testing" in our manuscript. We fully agree with your view. The "dual" nature of our system is indeed mainly reflected in the parallel amplification and detection capabilities of two target genes (*uidA* and *rcsA*), rather than achieving physical separation or synchronous differentiation of signals within a single reaction tube. We sincerely apologize for any misunderstandings that may have been caused by our unclear expression.

In the current experimental design, we will conduct the CRISPR/Cas12a reactions for *uidA* (*E. coli*) and *rcsA* (*K. pneumoniae*) in different reaction tubes. This is mainly based on the following two key considerations:

Ensure the specificity and reliability of the detection: As you know, the CRISPR/Cas12a system will perform non-specific cleavage on any single-stranded DNA in the system after activation. If guide RNAs (gRNAs) and reporter molecules targeting two different pathogens are placed in the same reaction tube, the "trans-cutting" activity of Cas12a triggered by one target will indiscriminately degrade all reporter molecules, leading to cross-interference and making it impossible to accurately determine whether the other pathogen exists. To completely avoid this potential "crosstalk" and ensure the independence and accuracy of each test result, we chose a physically separated reaction system.

Maximize the sensitivity and practicality of the method: Despite being responsible for separate detection, our method still achieves rapid and highly sensitive detection of two pathogens on the same heating block, using the same thermal cycling program, and within the same time. This represents a significant improvement in speed and operational simplicity compared to the traditional PCR testing that requires two independent experiments to complete. This "multiple targeting and parallel processing" strategy is the optimal choice after balancing detection specificity, sensitivity and operational efficiency.

To clearly illustrate this point in the manuscript, we will make the following revisions based on your suggestions. In the method section, we have added the subsection "2.3 Establishment of a Dual RPA-CRISPR/Cas12a Detection Method".

“To achieve efficient synchronous detection of *E. coli* and *K. pneumoniae*, we have developed a detection system based on dual RPA and CRISPR/Cas12a. During the nucleic acid amplification stage, we designed primers targeting the *E. coli* conserved gene *uidA* and the *K. pneumoniae*-specific gene *rcsA*, respectively, and simultaneously amplified the two targets in the same RPA reaction tube. However, due to the non-specific activity of the activated Cas12a protein in cutting single-stranded DNA, to prevent crRNAs targeting different pathogens from interfering with each other in the same reaction system, in the CRISPR/Cas12a

detection step, we amplified the RPA product. Incubate with specific crRNA and Cas12a protein, respectively, in independent reaction tubes. Ultimately, by real-time monitoring of the fluorescence intensity of each reaction tube, specific identification and determination of the two pathogens were achieved". (180-191)

Thank you again for helping us enhance the rigor and clarity of our paper. We hope the above explanations and the upcoming revisions to the manuscript can fully answer your questions.

2. In lines 168-171, the differences between crRNA1 and crRNA2 are not clearly described. Please explain their respective roles and design rationale in the main text.

Dear Reviewer:

Thank you very much for your careful review of our manuscript and the valuable suggestions you have provided. The issue you pointed out regarding the insufficiently clear description of crRNA1 and crRNA2 is very pertinent. We apologize for this oversight and have supplemented and revised the corresponding parts of the manuscript to more clearly explain their respective functions and design principles.

We have supplemented and revised subsection 3.2 Screening of RPA Primers and crRNA in the manuscript as follows:

"We designed two crRNAs, respectively for the *uidA* gene and the *rcsA* gene (named *uidA*-crRNA1/crRNA2 and *rcsA*-crRNA1/crRNA2). The design of crRNA follows its fundamental principle: each crRNA contains a conserved skeleton sequence (21 nt) for binding to the Cas12a protein and a variable interval sequence (20 nt). This spacer sequence binds to a specific homologous sequence on the target DNA (i.e., the *uidA* or *rcsA* gene) through the principle of base complementary pairing, thereby guiding the Cas12a protein to precisely locate at the target site and cleave the target DNA and the single-stranded DNA fluorescent reporter probe in the system. In the design process, we ensured that the spacer sequence of each crRNA has a high degree of gene specificity and targets the conserved regions of the genes to maximize the reliability and sensitivity of the detection. Subsequently, we evaluated the cutting efficiency of each crRNA through in vitro CRISPR/Cas12a fluorescence detection experiments. As shown in Figures 2C and 2D, compared with the negative control, all four crRNAs were able to successfully guide Cas12a to cut the fluorescence reporter probe and generate fluorescence signals. However, the cutting reactions guided by *uidA*-crRNA2 and *rcsA*-crRNA2 exhibited significantly higher fluorescence signal intensity and initial reaction rate, indicating that they have higher cutting efficiency. Therefore, based on their superior cutting dynamics performance, we selected *uidA*-crRNA2 and *rcsA*-crRNA2 for all subsequent experiments.

3. In the section "Optimization of the Dual RPA-CRISPR/Cas12a Method" (lines 176-192), it is unclear what criteria were used to compare band quality across different conditions. Please specify how amplification performance was evaluated (e.g., band intensity, specificity, absence of smearing).

Dear Reviewer:

Thank you very much for your valuable suggestions. The issue you pointed out regarding the insufficiently clear description of the evaluation criteria for the RPA condition optimization part is extremely crucial. We apologize for this oversight and have revised the section "Optimization of Dual-Target RPA-CRISPR/Cas12a Method"

in the manuscript based on your suggestions to clearly explain the specific criteria we use to evaluate the amplification performance of RPA under different conditions.

We have added evaluation criteria to the manuscript in 2.6 Optimization of a Dual-RPA-CRISPR/Cas12 Method:

“The amplification performance was evaluated based on three main criteria: (i) The strength of the amplification band on the agarose gel, indicating the yield; (ii) Specificity, indicated by a clear band on the expected molecular size of each target; (iii) Clarity of the electrophoresis background, minimal application or non-specific products”. (226-229)

Add an evaluation criterion in the CRISPR optimization section: “Select the optimal concentration of each component based on the criterion for generating the highest fluorescence intensity, which corresponds to the maximum cleavage activity.”(239-241)

4. In lines 201-215, optimization was performed by varying one factor at a time. Since this approach does not necessarily identify the optimal overall condition, please provide a direct comparison between the baseline (unchanged) condition and the conditions obtained after optimizing each individual factor.

Dear Reviewer:

Thank you for your valuable suggestions. We fully agree with your point of view that single-factor optimization methods have limitations and may not be able to directly determine the global optimal condition. To clearly respond to your request, we have made an intuitive and quantitative direct comparison between the baseline (conditions without changing any factors) and the results after each single-factor optimization.

We made modifications to Figures 4 and 5 in the manuscript. This diagram systematically presents the following comparison:

Figure A: It verifies the necessity of the detection system. Only when all key components are present can a significantly higher specific signal than the background be generated, which provides an effective basis for subsequent optimization.

Figure B-D: Direct quantitative comparison between single-factor optimization and baseline. We optimize each of the three key factors (Cas12a, crRNA, ssDNA) one by one:

Comparison method: In each subgraph, we simultaneously present the signal values (RFU) of the experimental group and the baseline group at each concentration (bar chart).

Evaluation index: More importantly, we calculated and plotted the curve (line graph) of the key indicator, the signal/background ratio (S/N, that is, the RFU of the experimental group/baseline RFU) varying with concentration.

Conclusion and Discussion

These data clearly indicate that through single-factor optimization, each factor can significantly enhance the signal-to-noise ratio of the system (relative to the baseline) under specific conditions.

Meanwhile, the variation curve of the S/N ratio also visually reveals the optimal range of action for each factor. The optimal concentration should be selected by comprehensively considering the cost and detection efficiency.

The problem you pointed out is of vital importance to our research. This supplementary analysis not only directly addresses your concerns but also makes the logical chain of our work more complete. Thank you again from the bottom of our hearts for your meticulous and professional review.

5. For the clinical strain validation, the manuscript does not clearly describe how clinical samples were processed prior to the RPA-CRISPR/Cas12a assay. Please clarify the sample preparation workflow, including any extraction or pretreatment steps.

Dear reviewer:

We sincerely thank the reviewer for this valuable comment. We apologize for the lack of clarity regarding the clinical sample processing workflow in our original manuscript. We have now thoroughly revised and supplemented the "Materials and Methods" section to provide a detailed description.

The detailed workflow for genomic DNA extraction from clinical bacterial strains is as follows:

Bacterial Culture Harvesting: Clinically isolated bacterial strains were inoculated into appropriate liquid medium and cultured under optimal conditions to the late-logarithmic growth phase. Then, 1-5 mL of the bacterial culture was centrifuged at 10,000 rpm for 1 min at room temperature. The supernatant was completely removed to obtain the bacterial pellet.

Cell Lysis: The pellet was resuspended by vortexing with 200 μ L of Buffer GA. Then, 20 μ L of Proteinase K was added and mixed, followed by the addition of 220 μ L of Buffer GB. The mixture was vortexed for 15 sec and incubated at 70°C for 10 min until the solution became clear, ensuring complete bacterial cell lysis.

DNA Binding and Purification: Then, 220 μ L of absolute ethanol was added to the lysate and mixed thoroughly. The entire mixture was transferred to the Spin Column CB3 and centrifuged at 12,000 rpm for 30 sec, allowing the genomic DNA to bind specifically to the silica membrane. After discarding the flow-through, the column was washed sequentially with 500 μ L of Buffer GD and 600 μ L of Buffer PW (with ethanol added as instructed). This washing step was performed twice to remove impurities such as proteins and salts.

DNA Elution: The spin column was air-dried at room temperature for several minutes to remove residual ethanol completely. It was then transferred to a clean microcentrifuge tube. Genomic DNA was eluted by adding 50-200 μ L of Elution Buffer TE to the center of the membrane, incubating at room temperature for 2-5 min, and centrifuging at 12,000 rpm for 2 min. The eluate containing the bacterial genomic DNA was collected.

In summary, genomic DNA extraction was performed strictly according to the manufacturer's instructions of the TIAN amp Bacteria DNA Kit. The extracted DNA was used directly as the template for the subsequent RPA-CRISPR/Cas12a assay, with 2 μ L used as input per reaction.

Revisions in the Manuscript:

The above-detailed sample preparation steps, including the kit information (TIANGEN, DP302), key operational parameters (centrifugation speed, temperature, duration), and final elution volume, have been integrated into the "2. Materials and Methods" section of the revised manuscript. (132-141) We believe these clarifications fully address your concern. Thank you again for your thorough and constructive review, which has significantly improved the clarity and rigor of our manuscript.

Minor Comments

1. Some references appear to be inappropriate or not well matched to the statements they support; please review and revise accordingly.

Dear reviewer:

We sincerely thank the reviewer for raising this important issue. We have conducted a thorough, line-by-line review of all citations and their corresponding statements in the manuscript. We fully agree that the accuracy and appropriateness of references are fundamental to the academic rigor of our work. We apologize for any oversights and have made comprehensive revisions accordingly.

Specifically, we have taken the following actions: We replaced references that were not the most direct or authoritative sources for the claims being made, ensuring each citation now provides strong and specific support for its associated statement. We also added key foundational literature to strengthen important general claims.

The main revisions include:

In the Introduction, the statement regarding the limitations of PCR (previously referenced as [8]) has been revised. We replaced the previous citation with a more appropriate and authoritative reference (Mackay IM, Arden KE, Nitsche A. 2002. Real-time PCR in virology. *Nucleic Acids Research*30:1292-1305) that extensively discusses the constraints of PCR in detection applications.

Also in the Introduction, we found that the example citing Zhang et al. [15] (a CRISPR sensor for circulating tumor cells) was less relevant to the paragraph's focus on pathogen detection. We have therefore removed this example to improve the focus and flow of the narrative.

In the Discussion, the broad statement about the significance of the CRISPR/Cas system in molecular diagnostics has been strengthened by adding the seminal, foundational papers in the field: Gootenberg, J. S., et al. (2017). Nucleic acid detection with CRISPR-Cas13a/C2c2. *Science* [15]; and Chen, J. S., et al. (2018). CRISPR-Cas12a target binding unleashes indiscriminate single-stranded DNase activity. *Science* [16]. This provides the necessary authoritative support.

We have also carefully reviewed and corrected other minor inaccuracies throughout the reference list to ensure consistency and precision. All these changes have been implemented in the revised manuscript. We believe these corrections have significantly improved the accuracy of the citations and the overall quality of the manuscript. Thank you again for the reviewer's insightful comment.

2. There are formatting inconsistencies in Figures 2 and 3 (e.g., "300bp" vs "300 bp"), as well as inconsistent font sizes in Figures 4 and 5. Overall, figures should be better standardized and organized.

Dear reviewer:

We sincerely thank the reviewer for this meticulous observation. We apologize for the inconsistencies in our figures. As suggested, we have conducted a thorough, figure-by-figure review of the entire manuscript to ensure complete standardization and organization.

The specific corrections we have implemented include:

Unit Formatting: We have standardized the notation for all units throughout all figures (Figures 1-5 and any supplementary figures) to include a space between the number and the unit (e.g., "300 bp", "50 °C").

Font Uniformity: We have unified the font type (Arial/Helvetica), size, and style (e.g., bold for titles) across all figure elements, including axis labels, tick labels, legends, and annotations in Figures 4, 5, and all other figures.

Overall Layout and Style: We have carefully adjusted the alignment, spacing, and

color schemes to ensure a consistent and professional visual presentation across all figures.

These comprehensive revisions have been applied to the high-resolution figure files in the revised manuscript. We believe these efforts have significantly improved the clarity, consistency, and overall quality of the figures.

Thank you again for this valuable feedback.

3. The term "screening" used in Figure 2 may be misleading given the limited number of conditions tested. Terms such as "comparison" or "evaluation" would be more appropriate.

Dear reviewer:

We sincerely thank the reviewer for this precise and constructive suggestion. We completely agree that the term "screening" could imply a broader scope of testing than what was performed. As suggested, we have replaced the term "screening" with the more accurate term "evaluation" throughout the manuscript to better reflect the nature of our work, which involved a direct comparison of a defined set of primers and crRNAs.

Specifically, we have made the following changes:

The title of Figure 2 has been revised from "Screening of RPA Primers and crRNA" to "Evaluation of RPA Primers and crRNA".

The corresponding text in the Results section (3.2) and the Figure 2 legend has been updated accordingly, replacing all instances of "screening" with "evaluation" (e.g., "Screening of primers for the *E. coli uidA* gene" is now "Evaluation of primers for the *E. coli uidA* gene").

We believe this change enhances the accuracy and clarity of our manuscript. Thank you again for this valuable correction.

4. Please ensure consistent formatting of bacteria names (italicized) throughout the manuscript and figure legends.

Dear reviewer:

We sincerely thank the reviewer for this meticulous comment. We completely agree that maintaining consistent formatting is crucial for the professionalism and readability of the manuscript.

We have conducted a thorough, word-by-word review of the entire manuscript, including the main text and all figure legends, to ensure that all bacterial genus and species names are consistently formatted in italics, in accordance with standard microbiological nomenclature.

Specifically, we have verified and corrected the formatting for the following and all other instances: *Escherichia coli* (*E. coli*), *Klebsiella pneumoniae* (*K. pneumoniae*) *Staphylococcus aureus* (*S. aureus*), *Pseudomonas aeruginosa* (*P. aeruginosa*)....

All these corrections have been meticulously implemented in the revised manuscript. We appreciate the reviewer's attention to detail, which has helped us improve the quality of our work.

5. Minor grammatical and typographical errors are present and should be corrected throughout the text.

Dear reviewer:

We sincerely thank the reviewer for this careful observation and valuable suggestion. We agree that precise language and formatting are essential for the clarity and professionalism of the manuscript.

We conducted a thorough line-by-line proofreading of the entire manuscript to identify and correct grammatical and punctuation errors and reformatted it. This comprehensive revision includes, but is not limited to:

Correcting subject-verb agreement and verb tenses. Ensuring the proper use of articles (a, an, the). Standardizing terminology and abbreviations for consistency. Checking and correcting punctuation (e.g., the use of commas and hyphens). Verifying the formatting of units (e.g., ensuring a space between numbers and units, as in "300 bp" and "37 °C").

All identified errors have been meticulously corrected in the revised version of the manuscript. We believe these efforts have significantly improved the readability and overall quality of the text. Thank you again for this helpful comment.

Re: Spectrum03598-25R1 (**Rapid and Simultaneous Detection of *Escherichia coli* and *Klebsiella pneumoniae*: A Novel Dual Recombinase Polymerase Amplification-Clustered Regularly Interspaced Short Palindromic Repeats/Cas12a Method**)

Dear Mrs. Lina Liang:

Your manuscript has been accepted, and I am forwarding it to the ASM production staff for publication. Your paper will first be checked to make sure all elements meet the technical requirements. ASM staff will contact you if anything needs to be revised before copyediting and production can begin. Otherwise, you will be notified when your proofs are ready to be viewed.

Sincerely,

Vittal Prakash Ponraj Ph.D
Editor
Microbiology Spectrum

Reviewer #1 (Comments for the Author):

The authors have substantially revised their manuscript by means of my and another reviewer's suggestions. Specifically, the clarity of the text has been significantly improved. All my concerns have been fully addressed, and I have no further questions.

Reviewer #2 (Comments for the Author):

The authors have carefully revised the manuscript and provided detailed responses to the comments. Overall, the revisions have improved the clarity and rigor of the manuscript.